# Amazon rainforest ecosystem exchange of $CO_2$ and $H_2O$ through turbulent understory ejections

Robbert P.J. Moonen[a], Getachew A. Adnew[d], Jordi Vilà-Guerau de Arellano[b], Oscar K. Hartogensis[b], David J. Bonell Fontas[a], Shujiro Komiya[c], Sam P. Jones[c], and Thomas Röckmann[a]

[a]Institute for Marine and Atmospheric Research, Utrecht University, Heidelberglaan 8, 3584CS, The Netherlands
[b]Meteorology and Air Quality Group, Wageningen University, Droevendaalsesteeg 4, 6708PB, The Netherlands
[c]Max Planck institute for Biogeochemistry, Hans-Knöll-Str. 10, 07745 Jena, Germany
[d]Department of Geosciences and Natural Recource Management, University of Copenhagen, Øster Voldgade 10, 1350 Copenhagen K, Denmark

**Correspondence:** Robbert Moonen (r.p.j.moonen@uu.nl)

**Abstract.** We investigate the role of short-term variability on the mean ecosystem exchange of carbon dioxide and water vapor. Specifically, we focus on quantifying how the intermittent turbulent exchange at the forest-atmosphere interface - characterized by sweeps, ejections and outward/inward interactions - contributes to the mean exchange. To this end we analyze observations of high-resolution (isotopic) flux measurements taken at 25 m above the forest canopy at the Amazon Tall Tower Observatory (ATTO) during the dry season. We identify short-term turbulent eddies that eject carbon dioxide and water vapor from the understory (0 - 15 m) into the atmosphere. The $H_2O$ ejected from the understory is shown to be depleted in deuterium ($^2H$) by 10 ‰ compared to $H_2O$ originating from the top canopy. We show that this matches the depleted water isotopic compositions found in understory leaf and soil samples. The diurnal cycle of the net ecosystem exchange (NEE) of $CO_2$ is presented as a function of the sweeping and ejection motions and understory flux contributions. Understory contributions average 1.4 % of NEE, but reach up to 20 %. In exploring the connection between intermittent canopy turbulence and cloud passages, we found a weak relationship (r = 0.027) between cloud passages and ejections, without a predominant influence of large clouds. These findings deepen our understanding of the gas exchange of the Amazon rainforest, which may ultimately allow decision makers to incorporate policies which can prevent the regions transition from a carbon sink to a source.

## 1 Introduction

Future climate is highly dependent on the response of the biosphere to the enhanced greenhouse effect. Both anthropogenic (e.g. deforestation) and climate feedback (e.g. drought) processes affect the capacity of the rainforest to take up carbon and recycle water across the South American continent (Rosan et al., 2024; Gatti et al., 2021). Direct measurements of the gas exchange can advance our understanding of the underlying processes and monitor the behavior of an ecosystem (Baldocchi, 2020).

It is difficult to capture the details of diurnal biosphere - atmosphere exchange in simplified mathematical representations due to highly non-linear exchange processes. One important example is the generally stably stratified layer of air below the

canopy crown, which is referred to as the understory (Pedruzo-Bagazgoitia et al., 2023; Machado et al., 2024). This layer has received little scientific attention as it is disconnected from the regular bulk turbulent exchange of $CO_2$ and $H_2O$ at the canopy crown which normally is facilitated by unstable atmospheric conditions during daytime. Combining insights from literature, we hypothesize the following mechanism by which the understory intermittently contributes to the net flux:

1. During characteristic convective days with developing cumulus clouds in the Amazon, a stable decoupled layer is formed under the canopy crown, where local emissions of $CO_2$, $H_2O$, and likely also VOC's, are trapped (Pedruzo-Bagazgoitia et al., 2023; Patton et al., 2016; Dupont et al., 2024; Fitzjarrald et al., 1988; Bannister et al., 2023; Thomas et al., 2008).

2. Increased vertical wind speeds (e.g. induced by shear or cloud dynamics) or shading by clouds break the stable inversion (Pedruzo-Bagazgoitia et al., 2023; Vilà-Guerau de Arellano et al., 2019, 2024; Lohou and Patton, 2014; Sikma et al., 2018).

3. As a result, air masses with understory characteristics are intermittently ejected up into the Atmospheric Boundary Layer (ABL) (Dupont et al., 2024; Patton et al., 2016).

Here, we investigate how to identify understory ejections in above-canopy, high frequency observations, what triggers these ejections, and whether the underlying $CO_2$ and $H_2O$ exchange processes below the canopy can be identified and quantified through measurements above the canopy. Conceptually, we follow the established framework from Shaw et al. (1983, (Shaw et al., 1983)), who combined fast fluctuations of the horizontal and vertical velocities u and w, to define the most energetic upward motions out of the canopy as ejections, and downward motions into the canopy as sweeps. We applied that framework to fluctuations of the chemical tracers $H_2O$ and $CO_2$ (Thomas et al., 2008). This allows us to identify and quantify for the first time the contribution of the understory ejections to the total Net Ecosystem Exchange (NEE) flux. Then we investigate pathways that may trigger the understory ejections, in particular the relationship between turbulent ejections and clouds.

## 2   Methods

We analyzed a comprehensive set of measurements obtained during a 2-week intensive campaign in the central Amazon rainforest at the ATTO tower (Amazon Tall Tower Observatory) in Brazil which was named CloudRoots Amazon22. Measurements encompassed the leaf scale, the canopy scale, and the ecosystem scale. Initial work from González-Armas et al. (2025), covers the link between the leaf and canopy scales. This manuscript is instead focused on the ecosystem scale, and how it links to known canopy scale processes. As the canopy level profiles presented in González-Armas et al. (2025) represent a local sample from a heterogeneous understory, they were not directly linked to the ecosystem scale.

Our measurement setup was installed on a balcony at 54 m above ground level ($\approx$30 m tall canopy). Central to the setup were an IRGASON EC-100 (Campbell Scientific, Logan, USA) and a LI-COR 7500 (from LI-COR Inc, Lincoln, U.S.A.), installed at 57 m height. These instruments continuously collected 20 Hz wind fields and concentrations of $CO_2$ and $H_2O$ . The data were evaluated using the processed and RAW output from EddyPro version 7.06 (Fratini and Mauder, 2014) (from LI-COR

Inc, Lincoln, U.S.A.). Data treatment includes double rotation of the wind field and WPL corrections (Wilczak et al., 2001;

Webb et al., 1980). An interquartile range (IQR) outlier filter was applied, using 2x the IQR to clean the isotopic compositions measured with the laser spectrometers (described below), and 3.5x the IQR for the mole fraction and wind field data (similar to Moonen et al. (2023)). The time series of the isotope analyzers and the eddy covariance system were aligned using the $H_2O$ or $CO_2$ mole fraction signals measured by both (Moonen et al., 2023). Here, the 20Hz wind and molefraction data from the EC were subsampled to match the 4Hz or 10Hz frequencies of the isotope sensors.

In addition to this backbone of high frequency wind and atmospheric composition data, two high flow rate laser spectrometers measured isotopic compositions of $H_2O$ (Picarro L2130-I) and $CO_2$ (Aerodyne TILDAS-CS)(Moonen et al., 2023). The analyzers were placed on the 54 m balcony in temperature-controlled enclosures, which were set to $35°C$ to be consistently warmer then the environment ($T_{max}$ $32°C$). Air was drawn to the analyzers via a 8 m, 1/2 ” heated copper inlet line positioned at 30 cm from the center of the anemometer tubing at high-flow rates ($> 20$ $Lmin^{-1}$) to ensure turbulent conditions. An

aluminum mesh inlet filter was used as an improvement to our previous field deployment (Moonen et al., 2023). Isotopic compositions are reported as $\delta$-values versus the international reference material VSMOW (Mook and Geyh, 2000). The calibration procedures are described in appendix A2.

## 2.1 Quadrant analysis

Quadrant analysis provides a way to visualize and quantify variability within a 30 min flux interval by studying the short time

deviations in wind and scalars from which the flux was calculated. The variations in vertical and horizontal wind $(u, w)$ or the heat flux $(w, T)$ are often investigated (Thomas and Foken, 2007). We instead looked at a quadrant analysis of $CO_2$ and $H_2O$ to best differentiate respiration and photosynthesis signals (Thomas et al., 2008) collocated with the wind perturbation signals. While similar features can be found in both $u, w$ and $CO_2, H_2O$ quadrants, $Q_1$ in $u, w$ space is not necessarily related to $Q_1$ in $CO_2, H_2O$ space (Fig. A5).

From this quadrant analysis of $H_2O$ and $CO_2$, quadrant-specific fluxes $F_{Qi}$ were derived for each 30 min interval. Each quadrant-flux can be linked to an exchange mode, possibly related to a specific process, and contributing to the total flux $F_{tot}$. Each resulting quadrant-flux was multiplied by the fraction of data points present in the respective quadrant ($\frac{n_{Qi}}{n_{tot}}$). The addition of the four partial fluxes equals the original 30 min flux (See Fig. 2 and Fig. A7).

$$F_{tot} = \sum_{i=1}^{4} F_{Qi} = \rho_m \sum_{i=1}^{4} \overline{w'_{Qi} \chi'_{Qi}} \frac{n_{Qi}}{n_{tot}} \tag{1}$$

Here, $\rho_m$ is the molar density of air in $mol\,m^{-3}$, and $\chi'$ is a time series of mole fraction fluctuations after Reynolds decomposition. Note that the fluctuations from the mean are based on the data from all four quadrants.

Thomas et al. (2008) also used the $CO_2$, $H_2O$ quadrant visualization, and identified data in $Q_1$ as the respiration contributions to the flux. They used a hyperbolic cutoff to isolate the ejection signals from the normalized detrended $CO_2$, $H_2O$ timeseries, which operates as follows.

$$lim < \frac{x'}{\sigma(x')} * \frac{y'}{\sigma(y')} \qquad (2)$$

Here, $lim$ indicates the value for the cutoff, and $x'$ and $y'$ represent the time series of the variable of interest. We found that a hyperbolic cutoff resulted in many data points from the bulk exchange mode to be falsely identified as ejections (Fig. 1). In contrast, the hyperbolic cutoff did work well for detecting wind sweeps and ejections in $u', w'$ space (Fig. A5).

In order to better identify understory ejections in $CO_2$', $H_2O$' space, we first fit an orthogonal distance regression (ODR) to the $H_2O$ and $CO_2$ anomalies. This fit generally follows the bulk exchange well, but is affected by anomalous turbulent features like ejections. To limit such effects, a second fit is applied to the data within a $2\,\sigma$ window of the first regression. This second fit always follows the bulk exchange more accurately. The points that deviate more than $2.5\,\sigma$ in the $Q_1$ direction from this second fit are considered ejections. For both thresholds, $\sigma$ is based on the the residuals in y. The blue dashed line in Fig. 1C shows the ejection identification threshold. The method takes the width of the distribution of the bulk exchange into account and thereby minimizes false positive ejection classifications when the distribution is wide. Separate bulk and ejection fluxes were also calculated following Eq. 2. Here, the division 'bulk' or 'ejection' was used instead of the division into $Q_{1,2,3,4}$ (Fig. 2).

## 2.2 Isotopic source compositions

We use the Miller-Tans method (Miller and Tans, 2003) to derive source compositions from 30 min intervals of 4Hz $H_2O$, $\delta^{18}O$-$H_2O$, and $\delta D$-$H_2O$ data. To limit errors in the source compositions of ejections, we only derived source compositions when at least 36 ejection data-points were present (from a total of n of 7200 in 30 min). The Miller-Tans method relies on fitting a line, of which the slope represents the isotopic source compositoin. We excluded cases where the standard error of this slope, following the error propagation described York (1968), was larger than 6 ‰. We also excluded physically unrealistic source compositions that fell outside the range -20 to 70 ‰. We speculate that in such cases, the mixing assumptions underlying the Miller-Tans were violated due the advection of air masses with different compositions.

The limited number of data points available for finding the ejection could force derived source compositions to zero due to the nature of some regression fitting algorithms. To prevent this, we used the York regression fitting algorithm recommended in Wehr and Saleska (2017), which incorporates the uncertainties in x and y, to apply a linear fit to the data (York, 1968). Moreover, we verified that our results are not biased by fitting, through confirming that a subset of bulk exchange data points, matching the ejections in number of points and y range, still resulted in similar isotopic source composition compared to all bulk data points (see appendix Fig. A3). The $CO_2$ isotope laser spectrometer was too unstable to derive source signatures from. Leaf and soil samples were analyzed for water stable isotopic composition following (Barbeta et al., 2019). In appendix A1, we detail our sampling approach and isotopic analysis method.

## 2.3 Cloud time shifting

The relation between cloud passages and ejections was investigated using lower frequency (10 s average) data from the 10:00 to 16:00 h interval. We define the "understory ejection intensity" as the fraction of time in the 10 s interval during which air parcels

were classified as understory ejections. We used an incoming Photosynthetic Active Radiation (PAR) sensor to determine when and for how long clouds passed. PAR and EC data were collected at 20Hz on one data logger, ensuring time synchronization. A "cloud intensity" variable was derived by calculating the percentage decrease of the measured PAR relative to a calculated clear sky PAR curve. Clouds that caused less than 10 % radiation dimming or lasted less than 20 s were excluded. The first interval during which a cloud was observed is referred to as the cloud onset. In total, 518 clouds were identified during the measurement period. 69 lasted for more than 4 min, which we refer to as "big clouds" (cloud size > 1 km given the $3.9 \text{ ms}^{-1}$ $U_{\text{avg, 57 m}}$ at ATTO).

To find a temporal relationship between clouds and ejections, we applied time-lagged cross correlation to the "understory ejection intensity" and "cloud intensity" variables. With a characteristic time scale of active cumulus clouds of around 20 min, and assuming cloud effects to be limited to 5 min before and after a cloud, we selected a 30 min time shift interval for this cross correlation (Romps et al., 2021). Coherent positive correlation coefficients should indicate if clouds trigger ejections, and the time of maximum cross correlation specifies at what time lag that relationship is strongest (Fig. 4).

We also determined the distribution of ejections around the onset of clouds. To prevent assigning one ejection to multiple clouds, all ejections were first assigned to the closest cloud onset in time. Then, we cumulatively added all ejections that occurred in a given time window around the start of each cloud. A consequence of analyzing at the closest ejection occurrence is that randomly generated cloud and ejection fields are correlated too (See Fig. 4). This issue is further discussed in results Sec. 3.4 and visualized by the synthetic data in Fig. 4C.

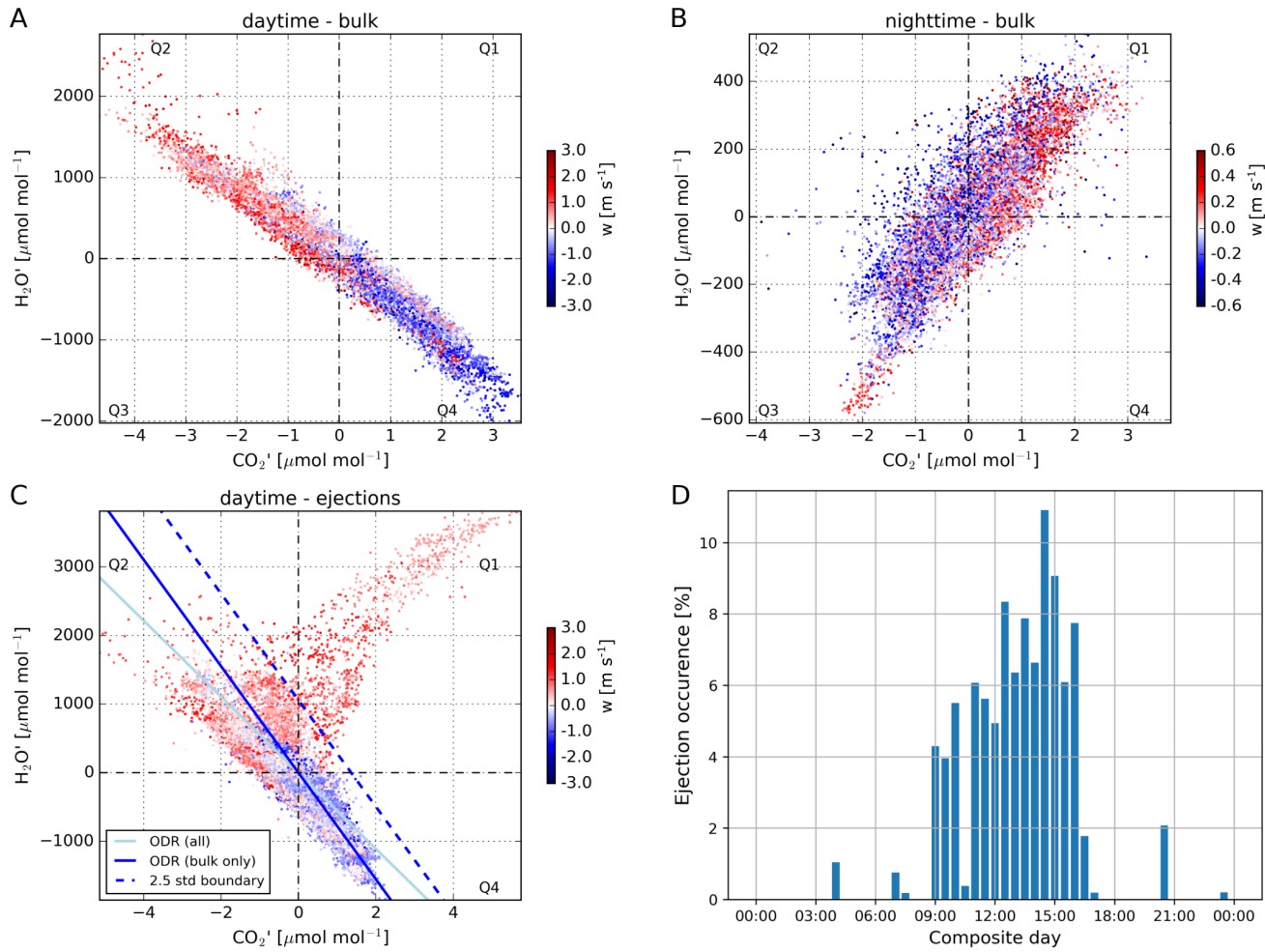

**Figure 1.** Panels A, B and C display 30 minute anomalies of the $H_2O$ and $CO_2$ mole fraction. Individual points are anomalies of the raw 10Hz data relative to the half-hour mean. The interval in Panel A runs from 14:00-14:30 LT on the 19[th] of August and represents the typical daytime bulk exchange. The interval in Panel B from 23:30-24:00 LT on the 20[th] of August and represents the typical nighttime bulk exchange. Panel C runs from 12:15-12:45 LT on August 18[th] and represents a daytime bulk exchange interval with an understory ejection in $Q_1$. The fitted lines are part of the automated ejection detection algorithm detailed in Sect. 2.1. Data points above the blue dashed line, that have positive w and occur in a streak of at least 10 successive points, are considered understory ejections. Panel D shows how these ejections are distributed over the day during the campaign. Here, the total duration of all ejections during the campaign was binned in 30 min intervals, and normalized to 100 %.

## 3 Results

### 3.1 CO$_2$ and H$_2$O turbulent transport: quadrant decomposition

Figure 1, shows the anomalies of CO$_2$ (x-axis) and H$_2$O (y-axis) associated with the bulk turbulent exchange, measured in the Atmospheric Boundary Layer (ABL). Each data point in panels A,B, and C is the deviation of a 10Hz observation relative to the 30 min mean, color-coded by the associated instantaneous vertical wind speed. Panel A shows a representative daytime 30 min interval (convective turbulent conditions with scatter clouds), where a negative correlation between H$_2$O and CO$_2$ is observed. Air parcels with upward motions shown in red are characterized by anti-correlated reduced mole fractions of CO$_2$ and increased H$_2$O mole fractions resulting from the opposing CO$_2$ assimilation and transpiration flux directions. Downward motions in Q$_4$ instead originate from the atmosphere above. See Fig. A5 in the appendix for a $u' - w'$ plot as defined by Shaw et al. (1983).

Panel B shows a 30 min nighttime example (stable stratification above the canopy, probably mechanical turbulence) in which we find a positive correlation between H$_2$O and CO$_2$. The pattern is a consequence of the combined CO$_2$ soil and plant respiration and evapotranspiration that feeds both CO$_2$ and H$_2$O into the atmospheric background shown in Q$_3$. Vertical motions during the night are weaker compared to those during the day, with maximal 30 min perturbations in $w'$ ranging from from 0 to 0.5 ms$^{-1}$.

Panel C shows another daytime example like in panel A. Here, however, a prominent streak of data-points deviating from the bulk trend is observed in Q$_1$. This upward-moving (red color) air mass must originate from a humid and respiration-dominated part of the biosphere, which can only come from the shaded understory. Therefore we classify these streaks as understory ejections. The $u' w'$ quadrant defined by Shaw et al. (1983) would categorize every understory ejection data points, and many others, as wind ejections (appendix Fig. A5). Using the understory separation as described in method Sect. 2.1, we isolated the occurrences of understory ejections based on the fitted dashed blue line. The ejection episodes identify the upward transport of understory air that is enriched in both CO$_2$ and H$_2$O, moves through the canopy crown, and into the ABL as observed at the 54 m measurement location. Most 30-minute flux intervals did not display understory ejections, and if they did, they generally contain fewer data-points and less extreme mole fraction anomalies compared to this example case. Detailed evaluation of the example in panel 3 shows that the streak of data points mostly represents two updraft events lasting for tens of seconds each (appendix Fig. A1).

Panel D shows the diurnal distribution of all observed ejections. While contributed varied, ejections were observed during all of the 13 days. They occur during most daylight hours and are most intense around 15:00 LT. This coincides with larger cumulus clouds developing according to previous studies, and confirmed by webcam footage captured on site (Tian et al., 2021). The coherence in the anomalies of CO$_2$ and H$_2$O reveals that these ejections around 15:00 LT are part of a second phase of ejections. The first phase (9:00-10:30 LT) is associated with a remnant of the morning transition, which is characterized by the flushing up of air that was trapped within the canopy during nocturnal stability (Dupont et al., 2024). The bulk of the flushing takes place from 7:00 to 9:00 LT, but the positive relationship between the H$_2$O and CO$_2$ anomalies prevents ejection events from being isolated then (Fig. 2). We therefore only observe ejections in phase one after photosynthesis has become

dominant around 9:00, and until the nighttime respiration residuals have mixed up at 10:30. Possibly the 'merging phase' of the residual layer from the day before and the newly developing convective boundary layer as described in Spiridonov and Ćurić (2021), is associated with the ejection-lean period at 10:30, between both phases. Wind direction analysis of all understory ejections suggests that ejections did not emerge under a preferential wind regime (appendix Fig. A2).

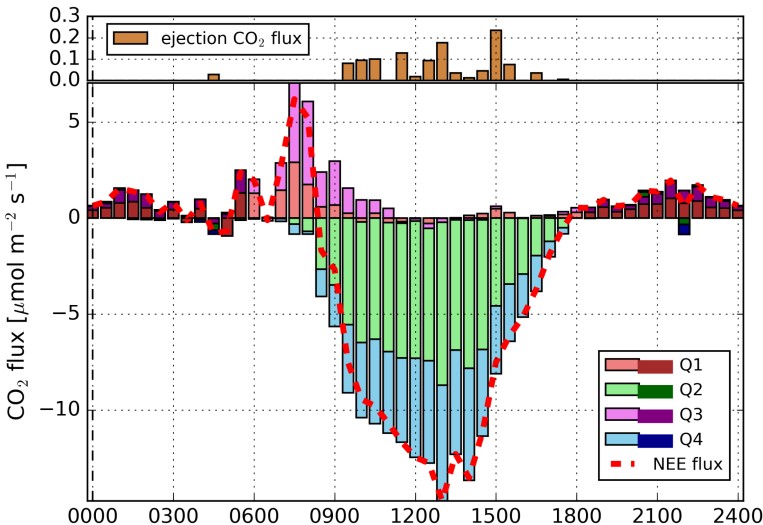

**Figure 2.** Composite diurnal cycle of the $CO_2$ NEE flux derived using quadrant analysis. The 30 min flux data is based on EC measurements taken from August 08 to 21 at 57 m. The quadrant analysis of fluxes is described in method Sect. 2.1. The dashed red line is the sum of all quadrant contributions, which equals the NEE flux. The darker colors indicate the nighttime hours. The ejection fluxes in the top panel were calculated analogously to the quadrant fluxes, using Eq. 1, only using the ODR 'ejection' and 'bulk' partitioning to specify the data subsets. Appendix Fig. A7 shows the quadrant analysis of $H_2O$ fluxes.

## 3.2 Quadrant analysis of $CO_2$ fluxes

Fig. 2 shows a 13-day average diurnal cycle of the 30 min $CO_2$ NEE flux. The colors indicate the contribution from each of the four $H_2O$ and $CO_2$ quadrants (methods Sect. 2.1). The top pane displays the vertical fluxes of the ejection events within each 30 min interval. From 09:00 to 17:00, flux contributions from ejections occur sporadically. Over the 13 day campaign period, ejection contributions to the flux summed to 1.4 % to the total NEE in absolute terms. Within 30 min flux intervals, ejection contributions regularly reached 20 %. On average, when ejections were observed in an interval, they contributed 3.8 % to the flux.

At 7:30, NEE is most positive, meaning that there is bulk transport of $CO_2$ from the canopy towards the ABL. $Q_1$ and $Q_3$ contribute equally, where $Q_1$ represents upward transport of air influenced by accumulated respiration from the soil and the

entire canopy and $Q_3$ the downward transport of air masses with low $CO_2$ and $H_2O$ contents. Simultaneously, photosynthesis becomes active, counteracting the upward $CO_2$ flux (Vilà-Guerau de Arellano et al., 2024).

From 9:00 onward, $Q_2$ and $Q_4$ dominate the bulk transport because the photosynthesizing canopy now controls the gas exchange (Fig. 1A). Around noon, the $CO_2$ canopy uptake flux is largest. During that time, 12 % of data-points in the $CO_2$ - $H_2O$ space are data points anomalous from the bulk exchange because they fall into $Q_1$ and $Q_3$ (see Fig. 1A). This percentage de-
creases over the course of a day as increased mixing makes the boundary layer more uniform in state variables and atmospheric composition. During noon, the downward $CO_2$ flux caused by these anomalous bulk data points is partially counteracted by the upward understory ejection flux, which exists in $Q_1$ (Panel B). The result is a near 0 net flux in $Q_1$ and $Q_3$ during noon (Panel A).

At 15:00 LT, only 6 % of data points are anomalous, as the $H_2O$ and $CO_2$ relationship is tighter (well-mixed, see Fig. 7 and
$D_1$ in Vilà-Guerau de Arellano et al. (2024)). As a consequence we observe a net upward flux of $CO_2$ in $Q_1$ that correlates with the number of ejections (see Fig. 1D). The same can be seen in the $Q_1$ and ejection fluxes of the $H_2O$ quadrant fluxes (appendix Fig. A7).

During the evening transition (18:00 onward), the fluxes in all quadrants are close to 0, followed by a steady respiration signal in $Q_1$ and $Q_3$ throughout the night. Increased horizontal and vertical heterogeneity in combination with suppressed
vertical motions in the stable nocturnal boundary layer above the canopy result in a noisy flux signal.

### 3.3 Source identification

High resolution measurements of water vapor isotopic composition are used to compare the isotopic composition of the understory ejections with that of the bulk exchange (Griffis, 2013). Both $\delta D$ and $\delta^{18}O$ were measured but we focus on $\delta D$ because of a higher signal to noise ratio. Fig. 3A shows isotopic source signatures from a Miller-Tans plot analysis, which reveals a
consistently depleted source isotopic composition for the isolated understory ejections compared to the bulk exchange. Representative examples of Keeling and Miller Tans plots are shown in appendix Fig. A4 (Keeling, 1958; Miller and Tans, 2003). The understory source composition has a larger uncertainty due to the limited number of data points and the small range in $H_2O$ mole fraction, but this does not explain the difference (see method Sect. 2.2).

The source signature for bulk exchange reflects canopy-air interactions (Welp et al., 2012). Looking at the daily changes
in isotopic composition, a clear diurnal cycle emerges (Fig. 3). After sunrise at 6:03, the depleted night signature enriches, peaking at 8:30, most likely because of dew evaporation and leaf transpiration (Cernusak et al., 2016). A stable midday plateau ends post-sunset at 18:04, followed by a decline to more depleted night values, indicating water re-equilibration (Cernusak et al., 2016). Note that the uncertainty in the source signatures is large during nighttime. Still, events with more depleted source compositions occur at 23:00 and at 04:00 in this composite diurnal cycle, where the 4:00 event is possibly related to
dewfall (Li et al., 2023).

During the midday plateau (9-17 LT), sufficient ejections (Fig. 1D) allow for reliable determination of the understory source composition (Fig. 3A). Here, a consistent deuterium depletion in understory ejections is observed compared to the bulk exchange. Differences can reach up to -10 ‰ between 14:30 and 15:30. Fig. 3B shows the midday isotopic compositions of

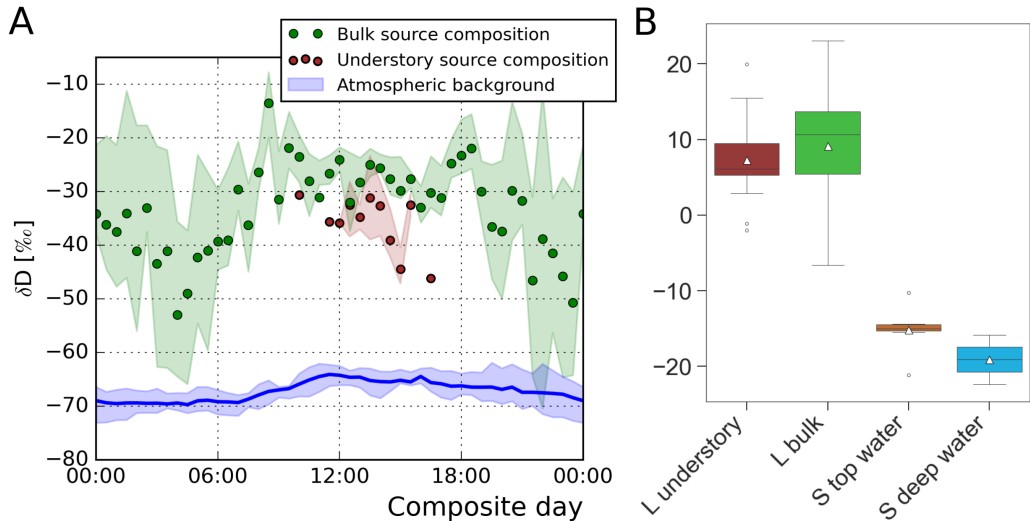

**Figure 3.** Diurnal $\delta D$ isotopic source signatures derived from Miller Tans plots. Each symbol represents a 30 min data interval averaged over the period from August 08 to 21 2022. Understory contributions (brown symbols) can only be calculated for the period with frequent ejections (Fig. 1D). Note that the emitted water vapor is always isotopically enriched relative the ambient isotopic composition (blue). Leaf (L) and soil (S) liquid water samples shown in panel B were collected over 3 days during the 13 day campaign (appendix A1). Only leaf samples collected between 11:00 and 17:00 were used. The triangles represent the mean.

leaf and soil samples taken at different heights/depths. We find that shaded understory leaves and top soil water from which evaporation takes place, are isotopically depleted compared to bulk sunlit leaves (predominant at the canopy top). This is in agreement with previous studies (Cernusak et al., 2016). An ANOVA test indicates that the depletion of understory leaves compared to bulk leaves is not significant for our sample (p=0.34). The Craig - Gordon model was used to determine the isotopic composition of the liquid water at the exchange site ($\delta_e$), as a function of the isotopic composition of atmospheric water vapor ($\delta_a$), the source water ($\delta_s$) of the plant, and environmental variables (appendix A3). In line with our understanding, it suggests that $\delta_e$ of leaves from the bulk canopy is 20 ‰ more enriched compared to the leaf water average derived from our sample (Cernusak et al., 2016). In addition, $\delta_e$ of understory leaves is depleted by 1.9 ‰ compared to the leaves from the bulk canopy. This supports the idea that understory leaf water is comparatively depleted too. The 10 ‰ Deuterium depletion of understory source air in Fig. 3 cannot qualitatively be explained by a $\approx 2$ ‰ depletion in leaves only. Liquid water in the soil is about 20 ‰ depleted compared to leaf water (Fig. 3B), and thus evaporated vapor from the soil should have a similar depletion of 20 ‰ compared to transpiration from leaves. We can thus expect that contributions from soil evaporation, which contribute to the understory source air, explain the 10 ‰ depletion we find.

Plant source water is the average of the water in the root zone, which can be sampled from the plant xylem (de Deurwaerder et al., 2020). We approximate the isotopic composition of the source water by taking the average of the deep soil samples (-20 ‰), and a sample of taken from a stream draining the local plateau (-27.4 ‰ at $\approx 70$ m lower elevation). This approximation

is supported by the fact that plant water uptake does not cause isotopic fractionation, and the knowledge that horizontal and vertical water isotopic gradients are limited in a rain forest with excess precipitation (Rothfuss and Javaux, 2017; Vega-Grau et al., 2021). We find that our plant source water (-23.7 ‰) closely resembles the daytime bulk ET source (-25 ‰), which confirms the isotopic transpiration equilibrium in plants (Yakir and da Silveira Lobo Sternberg, 2000)

### 3.4    Inferring Cloud-ejection relationships

Fig 4A shows a 80 min period where a period of ejections coincides with cloud passages. We find that understory ejections of $CO_2$ are strongly associated with wind ejections, which are displayed in the bottom panel. The ejection at 12:49 LT occurred in close proximity to a cloud onset. We expect ejections and clouds to be temporally correlated as clouds have the potential to break the in-canopy stability radiatively and/or dynamically (Fitzjarrald et al., 1988; Machado et al., 2024). Increased wind dynamics are caused by the large convective structures associated with clouds. Additionally, the cloud shading they provide

yields horizontal temperature gradient which trigger small-scale circulations above the canopy top (Horn et al., 2015). Our complete dataset comprises 13 days and contains 518 clouds and 379 ejections of different intensities.

     Fig. 4B shows the time-lagged cross correlation between the cloud intensity and ejection intensity (methods Sect. 2.3). Negative and positive x-coordinates represent the periods before and after the center of clouds passing, respectively. Note that the focus on the cloud center is not a choice, but a feature of the time-lagged cross correlation. At $x = 0$, which represents

ejections occurring during the center of a cloud passage, no correlation is observed with understory ejections. An increased probability of ejections is observed for negative time shifts, i.e., prior to the cloud center's arrival, with higher correlation coefficients for time shifts between -11 and -3 min. This suggests that relatively important dynamic and radiative effects occur before or at the start of the cloud triggering an ejections. We note that while the correlation is coherent, correlation coefficients are low ($r < 0.03$). An important cause is that many clouds and ejections occur in isolation (see Panel A). Another factor is that

we correlate an intermittent ejection time-series with a smooth, block shaped, cloud time series. While we find a temporal link between clouds and understory ejections, clouds of all characteristics (density and size) are included, making it unclear where ejections occur with respect to the cloud onset, or which cloud size contributes predominantly.

     A second approach was used to determine the ejection timing relative to cloud onsets (methods Sect. 2.3). In this approach we investigated the cloud starts instead of the centers and accumulated all ejection occurrences around those cloud starts. We

found that 23 % of clouds had one or more ejections in the 20 min window around their onset, while 66 % of ejections occurred within these same windows. This effect is caused by the clustered occurrences of ejections compared to the distributed nature of cloud fields (Romps et al., 2021). Fig. 4C only shows the ejection events in the 20 min window which were closest to the cloud onset, accumulated over 'all' or 'big' cloud cases.

     To visualize the bias associated with only selecting the closest, the blue bars show a synthetic ejection distribution. Here, we

applied a time shift of to the cloud field of at least one hour compared to the real measurement, and differing by 15 min between synthetic samples. Ultimately, each ejection gets collocated with 379 unrelated cloud fields. The synthetic data in Fig. 4C is the average histograms over these 379 runs. Only data from the 09:00 – 16:30 period, during which ejections are present, were used. In the 4 min window around the zero minute offset, the Cumulative ejection duration (y-axis) of the observed time

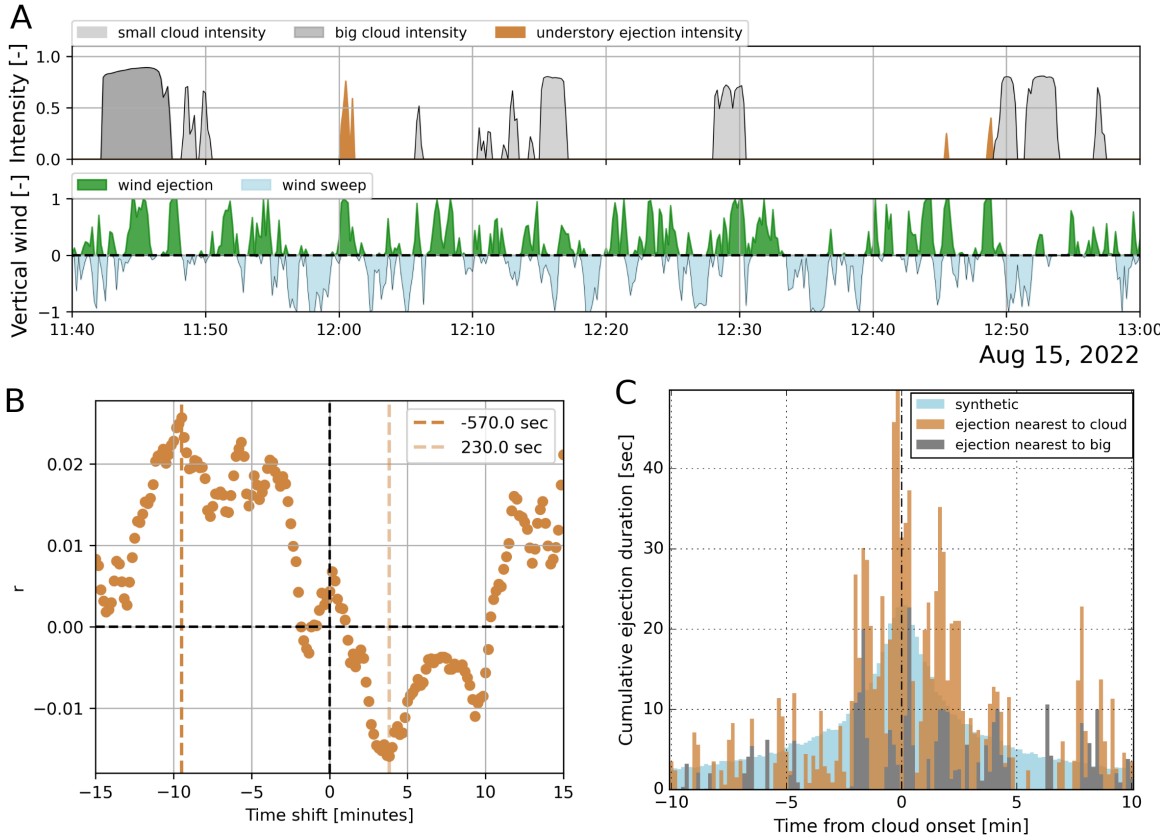

**Figure 4.** A: An example afternoon period from August 15 with many clouds, some understory ejections, and wind sweeps and ejections in the vertical wind panel. Clouds were identified using the difference between clear sky PAR and observed PAR. Big clouds were those that lasted for more than 4 min, which corresponds to a cloud size of about 1 km horizontally. Wind sweeps and all ejections are displayed as a fraction of time that an ejection occurred during the 10 s interval. B: The cloud - ejection relationship when time-shifting the data series by 10 s intervals. The vertical lines identify the maximum and minimum correlation between the 'cloud intensity' (small and big) and 'understory ejection intensity' variables. C: The cumulative distribution of ejections around cloud onsets using 10 s bins. Data ranged from 9:00 to 16:30 during the August $08^{th}$ to $21^{st}$ 2022 period.

series much exceeds the synthetic background, suggesting more ejections were present during cloud onsets then chance allows. While the pattern is noisy, an F-test confirms a significantly tighter distribution for 'all' clouds, than for the synthetic data (p = 0.02). This supports the idea that ejections preferentially take place near cloud onsets, and thus before the center of a cloud. The distribution for the 'big' cloud cases is not significantly tighter compared to the synthetic background.

## 4 Discussion

Our analysis demonstrates that fractions of trapped understory air regularly manage to escape through the canopy crowns towards the ABL (Fitzjarrald et al., 1988; Jacobs et al., 1992; Pedruzo-Bagazgoitia et al., 2023). These understory ejections present a small but measurable contribution to the NEE flux (1.4 %). Time averaged in canopy profile data from González-Armas et al. (2025) confirm that a disconnected understory layer was also present during our measurement campaign. In terms of composition, we use anomalous increases in $H_2O$ and $CO_2$ mole fractions to identify understory ejections. In turn, we find distinctly moist and $CO_2$ rich air coming from the understory. Interestingly, understory ejections were found to be associated with depleted water vapor isotopic compositions compared to vapor arising from bulk canopy transpiration. A depletion in the understory is supported by the difference in isotopic composition of water in sunlit leaves, shaded leaves, and the top soil (Fig. 3B), confirming previous studies (Yakir and da Silveira Lobo Sternberg, 2000; Cernusak et al., 2016). Besides, we show that the isotopic composition of the bulk transpiration source is of similar isotopic composition to the deep soil water.

While we verified the hypothesized relationship between clouds and ejections, this relationship was not as strong as anticipated. We regularly observe ejections which are not associated with clouds causing PAR shading. All understory ejections are linked to strong updrafts, which have been described here as wind ejections (appendix Fig. A5). Through the effect of wind shear, a negative correlation between horizontal winds (U) and understory ejections was also found when considering 10 s intervals. In appendix Fig. A6, we investigated if period with stronger wind gusts were related to periods with more ejections. In this way, we bypassed the dependence of wind and understory ejections on short time scales. When limiting our analysis to daytime intervals (09:00 - 16:30), we found no significant relationship between gusty and ejection rich period.

An advantage of our method for establishing cloud-ejection relationships is that we reduce the complexity of the 4D spatiotemporal observational space to a 1D point measurement, including detailed insight into the local atmospheric composition. This means that factors like wind direction and wind speed, which would strongly effect and complicate displaced measurements, drop out. As long as the wind direction remains similar over time and over the height of the boundary layer, an ejection triggered at some distance from the tower - and the cloud that caused it - are synchronously observed by the measurement station (Taylor's frozen-turbulence hypothesis). Some uncertainty is caused by our use of cloud shadows, which introduce unwanted horizontal displacements due to the varying zenith angle. Strong wind shear may also cause some uncertainty, as it will offset the arrival of cloud and composition data.

Finding a stronger causal mechanism triggering the observed understory ejections should have priority. To this end, better cloud data spatially collocated (<10 m) with wind, and composition data at a forest site are needed (Machado et al., 2024). More advanced cloud observational techniques like cloud radars or ceilometers would provide 3D spatiotemporal information, but these instruments are hardly portable, and are preferably installed on bare soil, limiting their use in forest environments. Possibly, webcams combined with cloud recognition analysis tools could provide an alternative.

With a more clear characterization of the trigger mechanism leading to understory ejections (Machado et al., 2024; Patton et al., 2016), the rapid development of canopy resolving large eddy simulation models could be validated (Pedruzo-Bagazgoitia

et al., 2023; Dupont et al., 2024). Moreover, multilayer canopy representations have shown to improve the next generation earth system models (Bonan et al., 2024).

## 5 Conclusions

We have used high temporal resolution measurements of $CO_2$ and $H_2O$ above the canopy of the Amazon rainforest, com-
305  bined with innovative analysis techniques, to investigate the intermittent interactions between the dry season rainforest and the dynamics of the atmospheric boundary layer. $CO_2$, $H_2O$ quadrant plots allowed us to dissect photosynthetic and respiration dominant exchange modes. Understory ejections, which transport air masses that have been in direct contact with the soil and lower canopy, can be found in turbulent quadrant $Q_1$ of such a plot. These intermittent, canopy penetrating motions constitute 1.4 % of the total $CO_2$ NEE flux, which matches the energy limited understory conditions. The isotopic composition of the
310  ejected understory air confirms that soil and plant water evaporation accumulated in the understory is depleted compared to water evaporating from the tree crowns. Clouds seem to act as a weak trigger for ejection events, but our findings are inconclusive about the dominant cause for the breaking of the canopy layer stability, which triggers ejections events. Future studies investigating understory ejections should attempt to find a causal trigger mechanism for understory ejections to occur, which can likely be achieved using tightly collocated (<10 m) wind, scalar, and advanced cloud observational data.

315  *Data availability.*  Data is available under open access with DOI: https://doi.org/10.6084/m9.figshare.27194964.v5

## Appendix A

### A1    Leaf and Soil sample analysis

Soil and leaf samples were collected systematically throughout daylight hours of August 14 and 15 2022. Soil samples were collected at 4 locations approximately 20 to 100 m from the base of the 80 m tower. Once a day between 07:00 and 10:00, soil from 0 - 10, 10 - 20, 40 - 50 and 80 - 90 cm was sampled using a corer at each of the four locations. Additionally a sample from 0 - 10 cm was collected with a trowel at the location closest to the tower at 07:00, 10:00, 14:00 and 16:00 on both days. All soil samples were stored in 20 ml, glass vials with positive insert caps. An additional core sample was also taken from the closest location at 14:00 on August 13 2022. Finally, a water sample was taken of the stream located at 70 m lower elevation, which drains the local plateau on which the measurement tower is situated.

Leaf samples were collected at approximately 07:00, 10:00, 14:00 and 16:00 on August 14 and 15 2022 from trees near the bottom of the tower ($\approx$ 2 - 3 m), those accessible from the middle ($\approx$ 12 - 15 m), and the top of the tower ($\approx$ 24 - 28 m). During each sampling leaves were collected from three trees. Similarly, additional samples were collected from all heights between 06:00 and 15:00 on August 12. All leaf samples were stored in gas-tight, 12 ml, glass vials. Samples were stored at 4 °C prior to further processing.

Soil and leaf water was extracted via cryogenic vacuum distillation West et al. (2006) at the Instituto National de Pesquisas da Amazônia (Manaus, Amazonas, Brasil). Briefly, the system consisted of four GL-18 glass extraction vessels and U-traps connected to a vacuum pump via a manifold and shut-off valve. Sufficient sample material to extract approximately 2 ml of water were weighed and transferred to extraction vessels. The samples were frozen in a dewar containing liquid nitrogen and the shut-off valve opened to evacuate the system. The shut-off valve was then closed, the extraction vessels transferred to a water bath and the U-traps moved into the liquid nitrogen. The water bath was set to approximately 90 °C and extracted water collected over the course of 2 h to ensure complete transfer between the sample and U-trap. Subsequently the U-traps were removed, capped and the collected water allowed to thaw before being transferred to 2 ml, plastic vials with positive insert caps. The extraction vessels and the samples they contained were then oven dried to verify there was no residual water.

The isotopic composition of the extracted water was measured on a Thermo Delta+XL IRMS coupled to a high-temperature conversion reactor (HTC) via a ConFlo III at the Max Planck Institute for Biogeochemistry, Jena, Germany Gehre et al. (2004). Calibration was made relative to two in-house water standards and a quality control tied to VSMOW-SLAP. Based on tests with 2 ml of water in place of sample soil and leaf material (n = 22), the average biases associated with the extraction system and analysis for $\delta^{18}$O and $\delta$D were - 0.27 $\pm$ 0.26 ‰ and -3.47 $\pm$ 1.77 ‰, respectively.

## A2  High frequency water vapor isotope calibration

The Picarro L-2130i was calibrated for $\delta$D and $\delta^{18}$O as described in Moonen et al. (2023). In short, we used one liquid water standard to apply a mole fraction calibration. Here, we determine the sensitivity of the instrument to variations in atmospheric water content on the isotopic signals. By adding the vaporized liquid water standard to a dry (synthetic air) stream in different proportions, we simulate such atmospheric water vapor changes, while keeping the inserted $\delta$D and $\delta^{18}$O compositions constant. When the measurements do return a varying $\delta$D and $\delta^{18}$O compositions, we correct for that.

A second calibration method was used to find the absolute offset between the measurement and the known composition of water standards. To this end, two liquid water standards from the IAEA were used, which spanned the range of observed atmospheric isotopic compositions IAEA (2017). One was w34 ($\delta$D = -189.48 ‰ and $\delta^{18}$O = -24.78 ‰), and the other was w39 ($\delta$D = +25.44 ‰ and $\delta^{18}$O = +3.63 ‰). The calibration approach was a span calibration, with an $H_2O$ setpoint of 15000 ppm.

Given that the instrument is very stable over time we used the average of the initial and final calibrations at ATTO, which were performed using the IAEA standards. Calibrations during the measurement campaign were also performed, which we used to confirm the stability. As these were made using different water standards of which the composition was less certain, we did not use them in our analysis. For the $CO_2$ isotope analyzer (Aerodyne TILDAS-CS), we did need all of the 20+ additional calibrations made during the campaign. Even then, we were not able to satisfactorily calibrate the instrument. Thermal instability was the main cause of the instrument drift, which was not sufficiently eliminated by Anodyne's field enclosure.

## A3 Craig - Gordon model

The Craig - Gordon model, first suggested by Craig and Gordon (1965), allows for the liquid water isotopic composition at the evaporation sites within leaves to be estimated. The enrichment of liquid water at the exchange site, resulting from faster evaporation of light isotopologues, is related to the water isotopic composition found in leaf samples, shown in Fig. 3B) (Cernusak et al., 2016). In addition, the source isotopic composition of transpiration is dependent on the exchange site water isotopic composition. Generally, variants of the Craig - Gordon model are used which assume an isotopic steady state between the source (xylem) water, and the isotopic composition of transpiration. Farquhar et al. (2007) provides an updated formulation of the Craig - Gordon model, which we implemented and define below.

$$\delta_e = \alpha^+ [\alpha_k(\delta_s + 1)(1 - \frac{w_a}{w_i}) + (\delta_v + 1)\frac{w_a}{w_i}] - 1 \tag{A1}$$

As we did not have access to leaf temperature measurements, we approximated $\frac{w_a}{w_i}$ with the relative humidity ($RH$), which is dependent on the air temperature instead. We combine data from our work with eco-physiology and profile measurements from our colleagues, which were also taken during the CloudRoots Amazon22 campaign (González-Armas et al. (2025), Fig. 2 and 3). A 14:00 case from a full-campaign data composite was analyzed. The key variables derived from either the data composite or the literature are displayed in the table below. We find that the exchange site liquid water ($\delta D_e$) is 1.9‰ more enriched in leaves at the canopy top compared to in the understory.

| Variable | Canopy top (bulk, ∼25m) | Understory (∼5m) |
|---|---|---|
| $\delta D_a$, atmospheric vapor | -65 ‰ | -65 ‰ |
| $\delta D_s$, xylem water approximated from mean of deep soil and runoff water samples | -24 ‰ | -24 ‰ |
| $RH$, relative humidity (in canopy profiles) | 0.58 | 0.65 |
| $T_{leaf}$ (air temp in canopy profiles) | 33 °C | 30 °C |
| $R_s$ (measured stomatal resistance) | 136 s m$^{-1}$ | 512 s m$^{-1}$ |
| $R_b$ (modelled boundary layer resistance, dependent on U, following Bonan (2002)) | 77 s m$^{-1}$ | 245 s m$^{-1}$ |
| $\alpha^+$ (liquid to water fractionation, following Horita and Wesolowski (1994)) | 1.0706 | 1.0735 |
| $\alpha_k$ (kinetic fractionation, following Farquhar et al. (1989)) | 1.0221 | 1.0224 |
| $\delta D_e$, exchange site liquid water (from Eq. A1) | 29.2 ‰ | 27.3 ‰ |
| $\delta D_L$, leaf sample liquid water (avg, see Fig. 3B) | 9.6 ‰ | 7.6 ‰ |

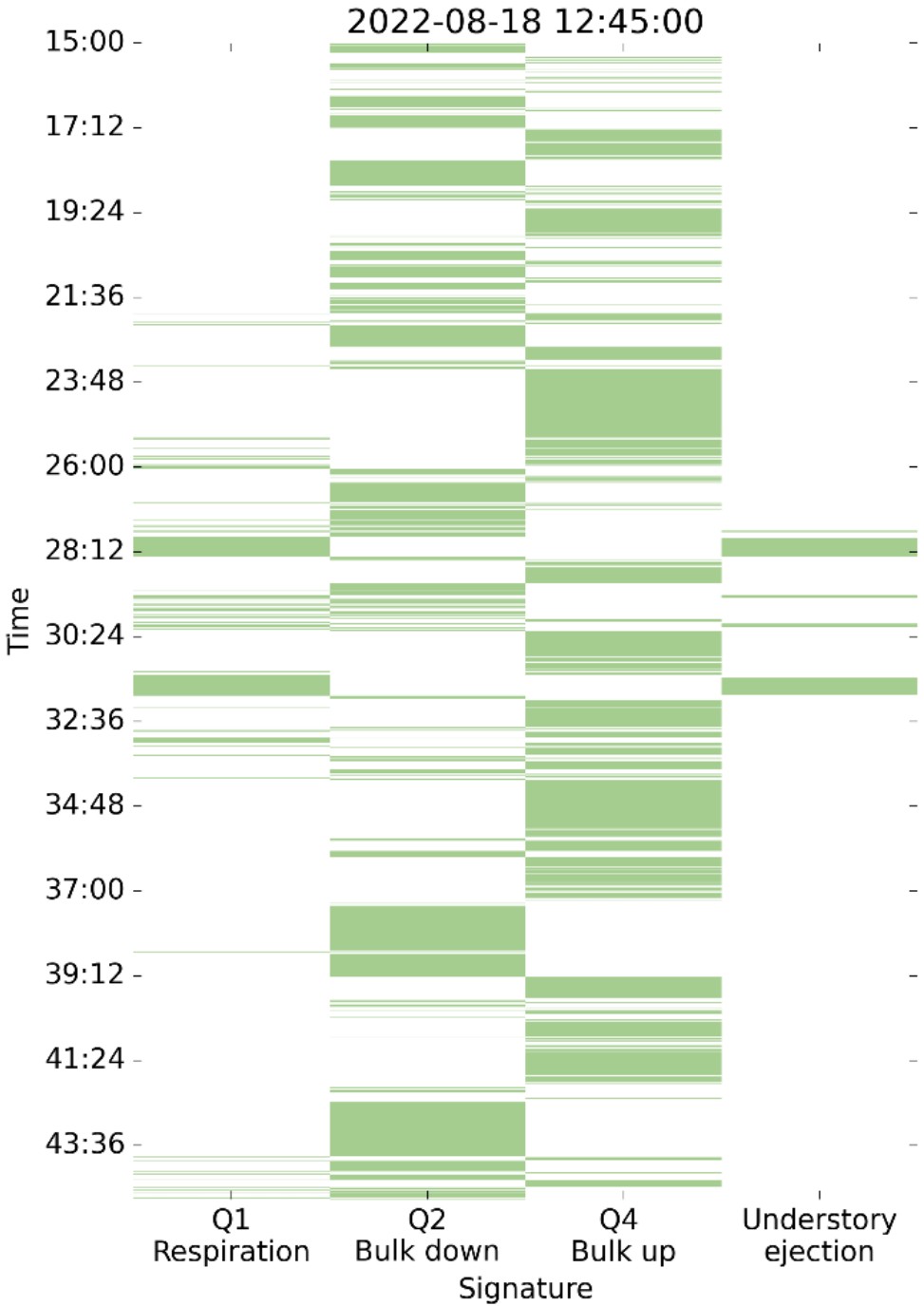

**Figure A1.** The temporal evolution of the quadrant contributions from Fig. 1C of the main manuscript. The green colors specify moments when data-points were part of a specific quadrant, or part of an ejection streak. Note that ejections predominantly observed in Q1.

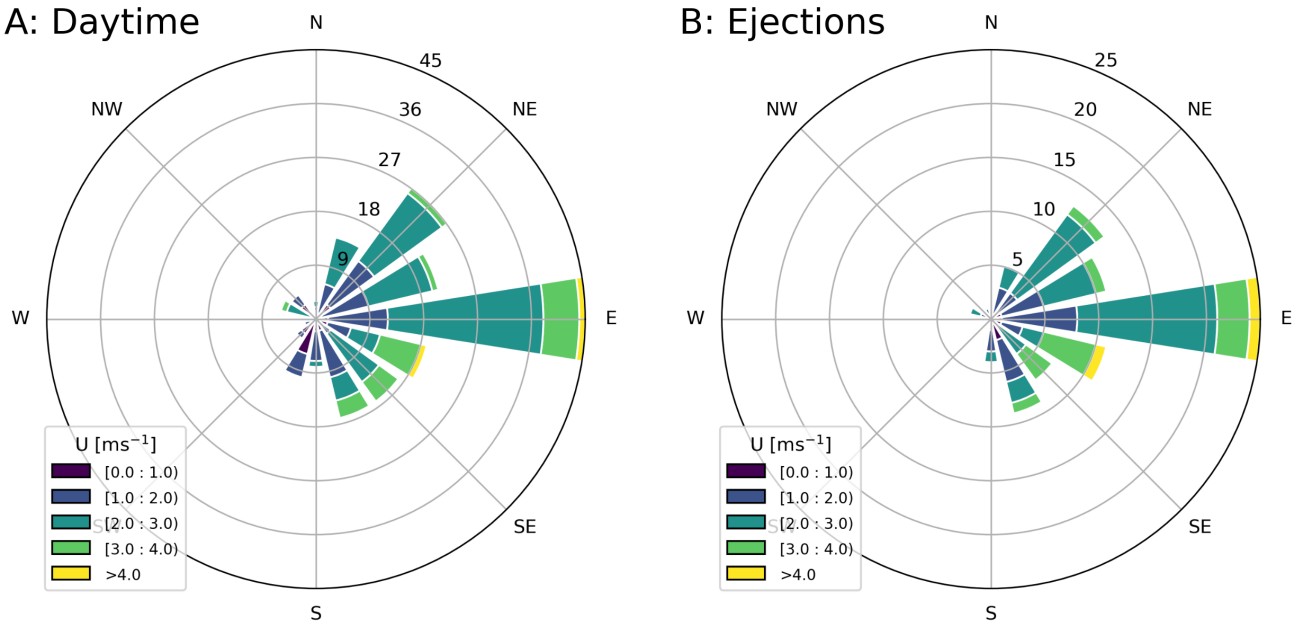

**Figure A2.** Wind roses of wind speeds and directions. In A; all 30 minute averaged wind directions observed between 9:00 and 16:30 are plotted, while in B; a subset of 30 minute averaged wind directions, from intervals in which ejections were observed, is plotted. The numbers on the diagonal indicate the number of data-points in each wind direction bin. The result suggests it is unlikely that a point source emission caused the ejections we observed, and that instead it is a physical feature linked to tall forest ecosystems.

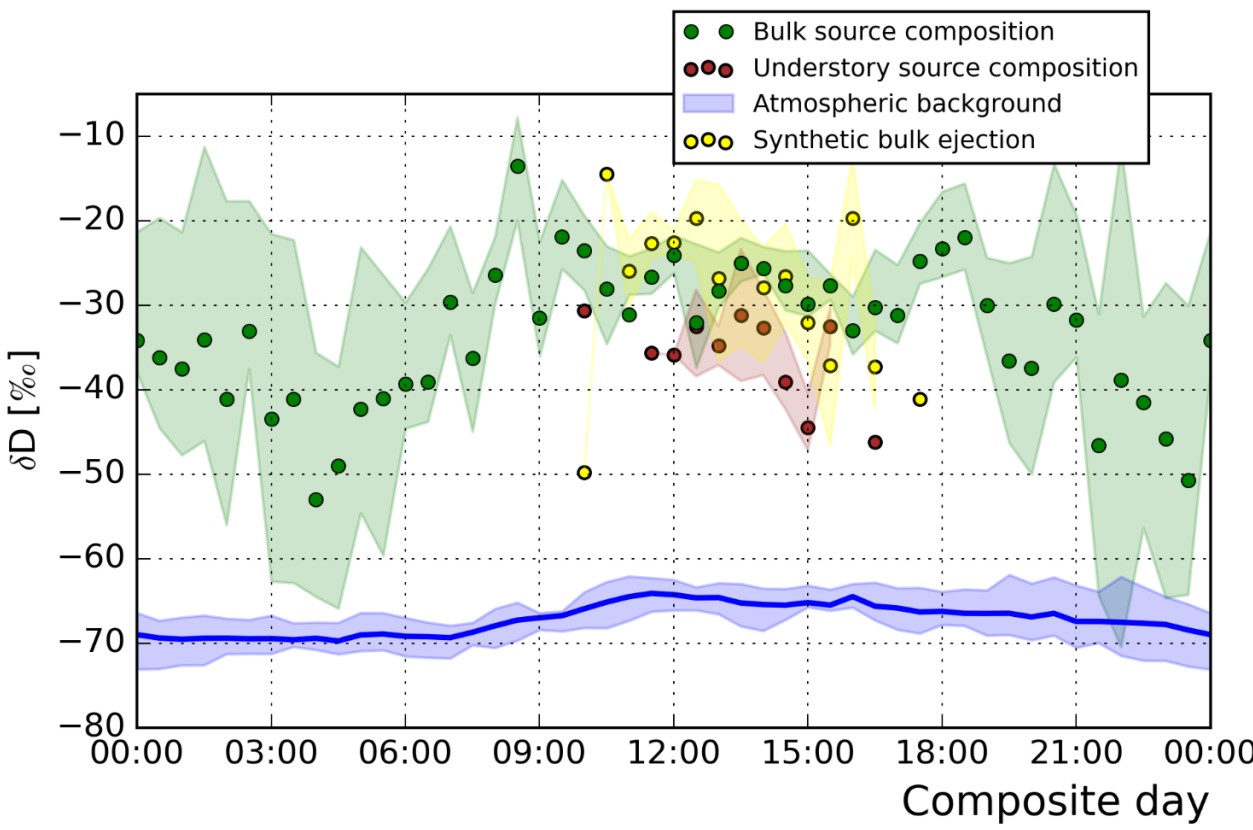

**Figure A3.** Source compositions derived using Miller-Tans plots. In addition to Fig. 3 of the main text, in which the bulk and understory source compositions are shown, a source composition was also derived for a subset of data points from the bulk source that spanned only 25 % of the $H_2O$ range, and included as few data points as are present in small ejections (yellow marks, $n$=36). The result indicates that although noise is introduced, the 'synthetic bulk ejections' do not show a similarly depleted source composition as the understory ejections (see Sec. 2.2). The depletion in the understory ejection is thus not a feature of challenging fits with limited $n$, but a feature related to the understory $H_2O$ composition.

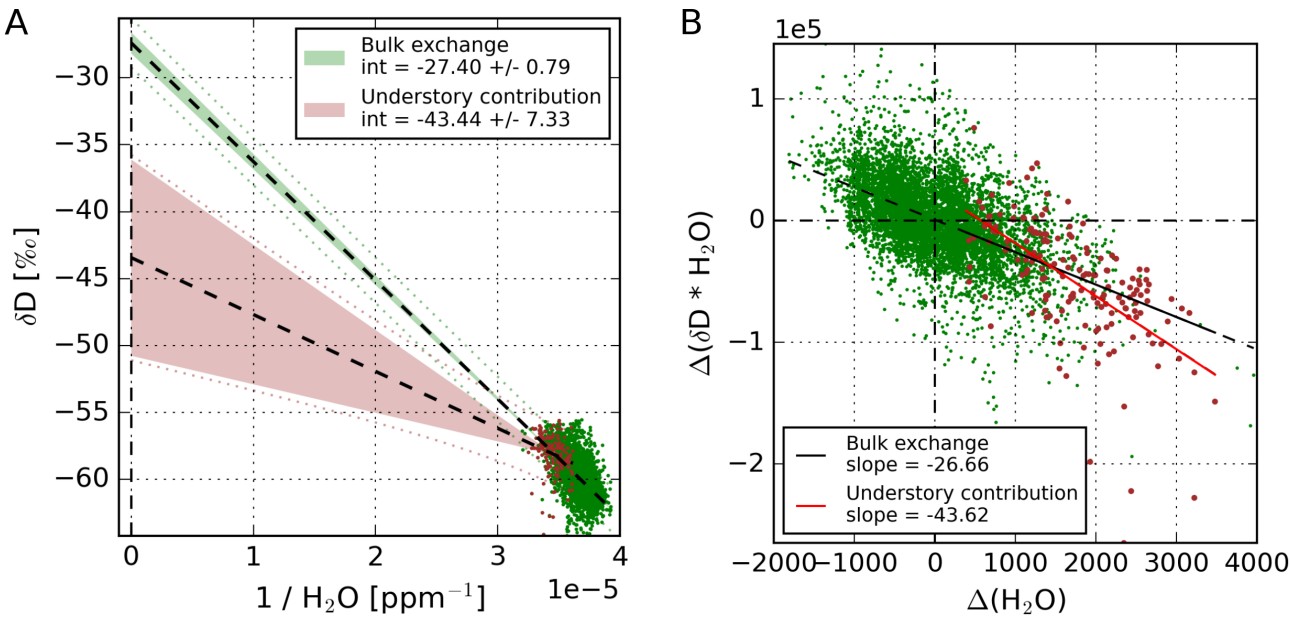

**Figure A4.** Example fits of the Keeling and Miller Tans methods for finding the source isotopic composition. One 30 minute example interval was used for both methods to show the differences in the resulting source compositions.

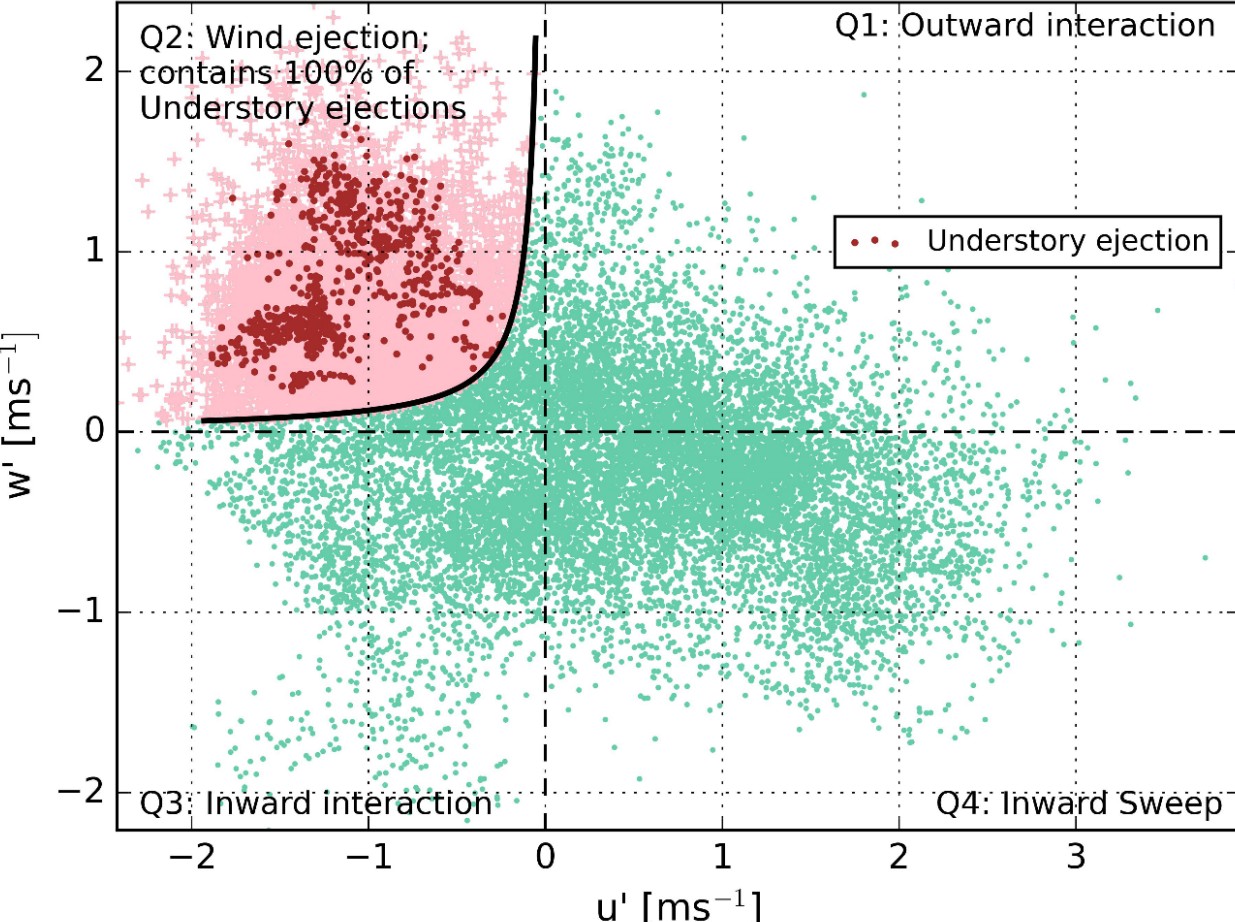

**Figure A5.** Quadrant analysis of the momentum flux. The black curve indicates the hyperbolic cutoff $-0.2 = \frac{w'}{\sigma(w')} * \frac{u'}{\sigma(u')}$, where $w' > 0$ (Eq. 2). Data that exceeds this threshold are shown in pink and are considered wind ejections as defined by Shaw Shaw et al. (1983). The example 30 minute interval is the same as is shown in Fig. 1C of the main text. The red dots indicate the data-points that were classified as understory ejections in Fig. 1C, $Q_1$. All understory ejection data-points are also wind ejections. Most wind ejections however are no understory ejections.

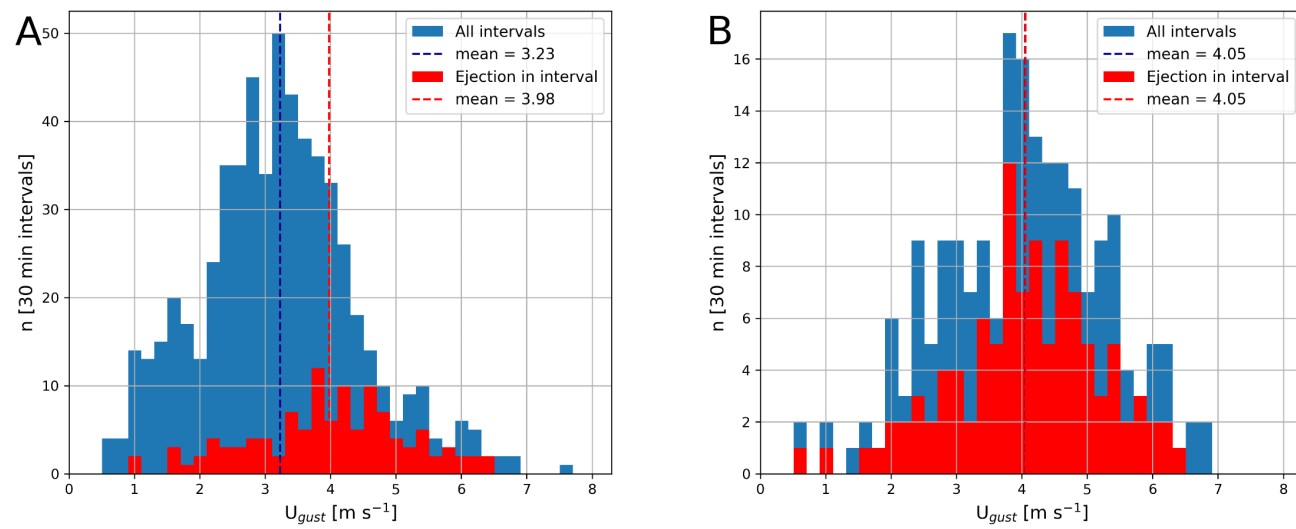

**Figure A6.** Distribution of the 30 min gust speeds. Panel A shows the result for both daytime and nighttime intervals. Panel B instead is limited to only daytime intervals (9:00 - 16:30). The gust speed was determined as being the $95^{\text{th}}$ percentile of 10 $sec$ U's over the 30 $min$ interval. In red, all 30 $min$ gust speeds during which ejections occurred are shown. A two sided t-test indicates that the red and blue populations are significantly different in Panel A with a $p$ of $4e^{-11}$. The populations in Panel B are not significantly different with a $p$ of 0.48.

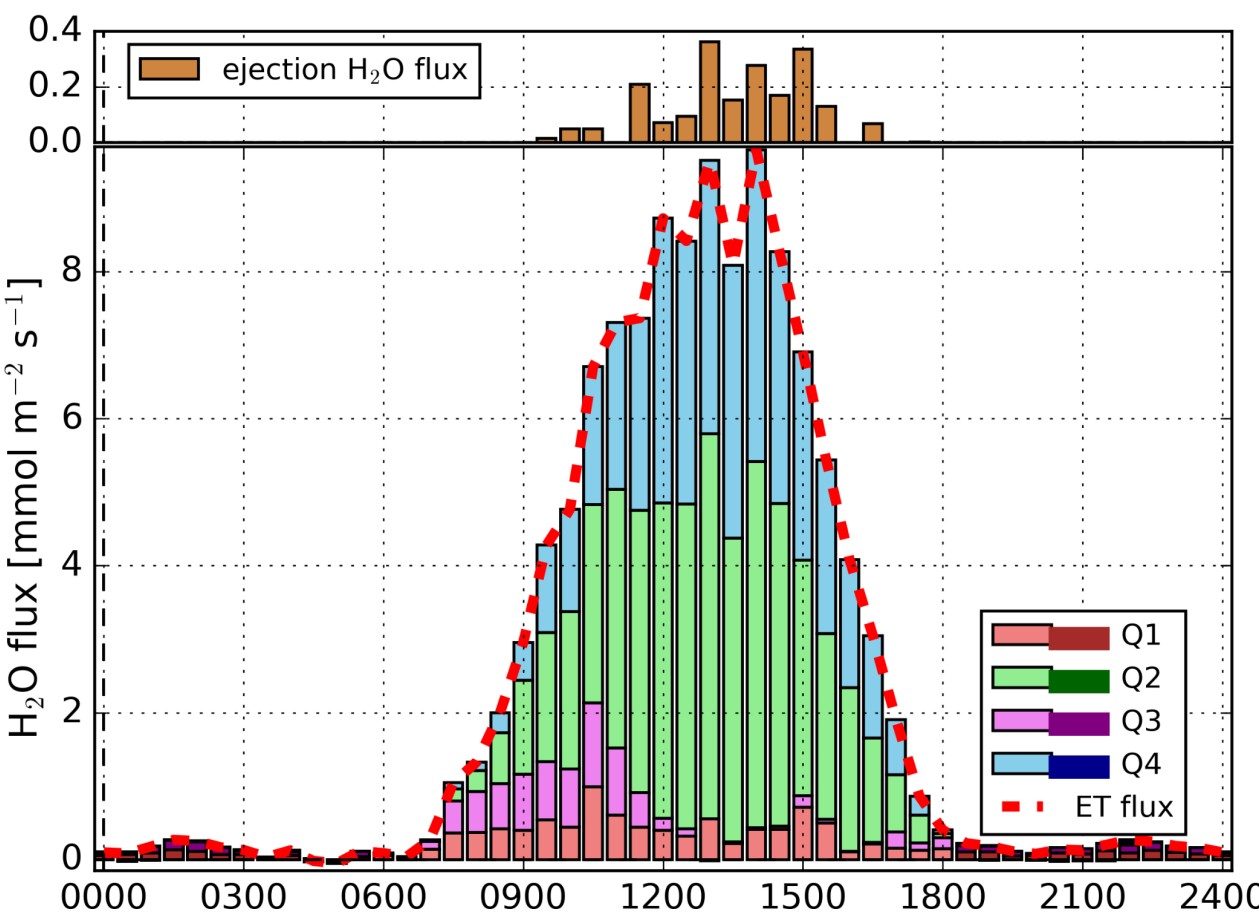

**Figure A7.** Composite diurnal cycle of the $H_2O$ flux derived using quadrant analysis. The 30 min flux data is based on EC measurements taken from August 08 to 21 at 57 m. The quadrant analysis of fluxes is described in method Sect. 2.1. The dashed red line is the sum of all quadrant contributions, which equals the $H_2O$ flux. The darker colors indicate the nighttime hours. The ejection fluxes in the top panel were calculated analogously to the quadrant fluxes, using Eq. 1, only using the ODR 'ejection' and 'bulk' partitioning to specify the data subsets.

*Author contributions.* The Utrecht and Wageningen teams realized the measurement setup and contributed to the interpretation of the measurements. R.P.J. Moonen was responsible for the data analysis and writing the manuscript. G.A. Adnew and the MPI Jena team collected and analyzed the leaf and soil samples. Corrections and suggestions for the manuscript were made by all authors.

*Competing interests.* The authors declare no conflicts of interest.

*Acknowledgements.* This work was funded by the CloudRoots project of the Dutch National Science Foundation NWO (grant number OCENW.KLEIN.407). The isotope instruments used in this study have been funded as part of the Ruisdael Observatory, a scientific research infrastructure project which is (partly) financed by the Dutch Research Council (NWO, grant number 184.034.015). We thank Marcel Portanger and Henk Snellen of Utrecht and Wageningen university for highly valuable technical support. Valmir, Davi, and Karl for their on
site support. The entire leaf and soil sample team; Jardison Valente Nunes, Maria Juliana de Melo Monte, Gloria Vieira Rodrigues, Amanda Rayane Damasceno Macambira, and Heike Geilmann.

The ATTO project has been funded by the Bundesministerium für Bildung und Forschung (BMBF Contracts 01LB1001A, 01LK1602B, and 01LK2101B), the Brazilian Ministério da Ciência, Tecnologia e Inovação (MCTI/FINEP Contract 01.11.01248.00), and the Max Planck Society. We also acknowledge the technical, logistic, and scientific support of the ATTO project by the Instituto Nacional de Pesquisas da
Amazônia (INPA), the Amazon State University (UEA), the Large-Scale Biosphere–Atmosphere Experiment (LBA), FAPEAM, the Reserva de Desenvolvimento Sustentável do Uatumã (SDS/CEUC/RDS-Uatumã), and the Max Planck Society.

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
