# Peer review of "Amazon rainforest ecosystem exchange of CO2 and H2O through turbulent understory ejections"

_EGUsphere, 2025_

## Author Comment (AC1)

Rebuttal of ACP manuscript; **Amazon rainforest ecosystem exchange of $CO_2$ and $H_2O$ through turbulent understory ejections**

We kindly thank the reviewers and the editor for the time they invested in improving the manuscript, and for their appreciative general remarks. We have addressed individual comments in blue, and will upload a track changes manuscript to indicate how we incorporated the feedback.

Comments from reviewer 1

*While I have a basic understanding of turbulent fluxes, I am not an expert in canopy issues. Overall I see this as an interesting paper illustrating aspects of turbulent fluxes through and above the Amazon rainforest canopy.*

*It would be good to see some statements and ideally profile information on mixing ratios of $CO_2$, $H_2O$ and $^2H_2O$ below, through and above the canopy in day and night time if any of these are available. The ejection concept seems to say that, above the canopy in daytime, there is generally a downward flux of $CO_2$ but when material is ejected from below the canopy, and turbulent fluctuations $CO_2'$ and $H_2O'$ are both > 0 it is implied that the sub canopy can be a $CO_2$ source while the canopy itself is a sink. It could be interesting to see fluxes below the canopy, as well as above, to see if there is absorption of sub-canopy $CO_2$ as it passes through the canopy.*

Reviewer 1 recognises the implications of our method and findings entirely correctly. Indeed, canopy mole fraction and / or flux profiles can help us understand the multiple in-canopy processes. While some profile data was available, we chose not to incorporate it in this manuscript as argued below. We realise that this argumentation is not fully laid out in the manuscript, while it would be insightful to the reader, as Reviewer 1 points out.

Firstly, stable stratification within the canopy causing increased concentrations of H2O and CO2 below the canopy top, due to daytime respiratory processes, are a relatively well-known feature of forest ecosystems (see references in the introduction at l. 26). They are the rule, and not an exception, which is why we did not feel the need to prove their presence in this manuscript.

The diurnal evolution of the vertical profiles of potential temperature, CO2 and H2O during our campaign is described in detail by González-Armas et al. (2025) . Note that these in-canopy profiles were collected at an ecosystem site at a distance of ~1km from the main tower at ATTO, from which we performed the flux measurements. DHO profile measurements are only available for a different time period, due to instrument down-time during the campaign period. Averaging over time windows, and multiple days (aggregate of 6 days), was necessary to retrieve these smooth profiles from point measurements. These profiles are representative

of a diurnal boundary layer transitioning from clear conditions to shallow convection around 10 LT.

[Figure]

*Figure A: From González-Armas et al. (2025, accepted preprint).*

In line with the literature and the expectations formulated in our introduction, we observe a strong stable temperature inversion in the canopy. In addition, we find a buildup of CO2 in the stable below-canopy layer, notably during daytime. Relative humidity is also highest in this stable layer. Note that RH is not a conserved variable, thus it is not optimal for indicating a buildup of water vapor. The same is true for the VPD shown in panel 2, which has a strong temperature dependence.

Importantly, profile data from a canopy with spatially heterogeneous stability, are of a very local horizontal scale (~100m2 footprint).  Our measurements of ejections (far) above canopy are representative for the ecosystem scale (~100,000m2 footprint). Therefore, local profiles cannot be linked directly to ecosystem scale ejections. This is especially true for the short timescales (min) we investigated.

Given these insights, the following changes were made to the manuscript:

- (L.45)

- o We refer to the work of (González-Armas et al., 2025)
  - o We specify the ecosystem scale of our measurements and indicate that direct links to profiles were not considered insightful.
- (L.240 (discussion))
  - o We mention the temporally averaged local profile data based on the six-day aggregate as presented in Gonzales-Armas et al (2025), which confirm a stable layer in the canopy with a visible respiratory signature.

*Nighttime seems less certain while Fig 2 shows a positive $CO_2$ flux in early morning. The discussion in Section 3, focussed on quadrant analyses from Figures 1 and 2 is very good.*

Indeed, at nighttime, the quadrant fluxes and net flux is uncertain. In the early morning, when the stability above the canopy is broken (7:00 – 10:00), a large positive CO2 flux is observed at 57 m. We understand that this comment was a confirmation of our findings and does not require adjustment.

*Links to cloud cover are explained and are interesting.*

*Detailed comments.*

*Abstract      Was the "the depleted water vapor isotopic compositions" measured in understory air? or just leaf and soil samples?*

This question points out a misleading statement in the original manuscript. No understory air samples were used in our analysis. We intended to refer to the aggregates of the isotopic compositions of the water samples taken from the soil and the leaves. This has now been clarified.

Changes:

- (L.8) Remove "vapor". Clarify that only liquid leaf and soil samples were taken.

*p3      The quadrant analysis in Figure 1, and especially 1c represents the important information presented here. As I read it there should be 36,000 points in each of Figs 1a,b,c (30 min each) but only 7200 (4Hz) isotopic composition measurements.*

Figures 1a,b,c, all contain 30 min, 10 Hz data, meaning 18000 data points are present. The subsampling from the 20 Hz raw wind field measurement was chosen to match the 10 Hz frequency of the CO2 isotope analyser. These data were ultimately not used in our analysis, due to uncertainties related to the instability of the instrument.

A second subsampled wind dataset was made to relate to the H2O isotope analyser, which had a 4Hz sampling rate. As a consequence, all H2O isotope - wind related

analyses are performed at 4Hz (thus 7200 data points). Still, we chose to display the ejection dynamics at the higher 10Hz frequency. We agree that the original 20Hz dataset might have been used as well to display the ejection dynamics, but we did (and do) not see enough added benefit over using the 10Hz dataset which we had already generated and analysed.

Changes:

- (L.58) The subsampling of EC data dependent on the isotope analyser measurement frequency is specified.

Note that our strategy for isolating the data points which we consider 'ejections', in which we combined the vertical velocity w, the orthogonal distance regression fit of $H_2O$ and $CO_2$, and the ejection duration (dt), was implemented for both the 4Hz and 10Hz datasets. For example, the 4Hz representation instead of the 10Hz representation of the data underlying Fig 1c was used as input for Fig 3a, since 4Hz water isotope data was required. While minor differences might occur between the ejection classifications in the 4Hz and the 10Hz representations, we believe these are insignificant for our analysis.

*I am not sure what is meant by "a hyperbolic isolation function".*

The "hyperbolic isolation function" is the function that we use to define when a signal is distinct from the bulk exchange mode. It is a recognised method for finding respiration signals in $CO_2$ – $H_2O$ quadrant analysis (Thomas et al., 2008). The function is shown in Fig. A5, in the caption of which we explicitly provide the function definition and specify the threshold. We acknowledge that "isolation" is not a clear term and changed the name to hyperbolic cut-off.

Changes:

- (L.74) Add function definition as eq. 2 to the explanation in Sec. 2.1
- (L.76) Add reference to Fig. A5 (u'w' plot) in Sec. 2.1
- Changed the name of the concept to "hyperbolic cutoff".

To explain the basics; the concept of the "hyperbolic isolation function" is taken directly from Thomas et al (2008). It provides a solution to an inherent problem with quadrants in turbulence data: Many data points are near the origin of the reference coordinates. When outliers from a dominant exchange mode are to be detected, we are instead interested in data points further from the origin. A hyperbolic function, namely, the multiplication of the normalised $H_2O$ and $CO_2$ mole fractions in one quadrant, provides an extra dimension to deny or accept data points near the origin. The threshold, 0.2 in the case of Fig. A5, specifies the sensitivity.

Note; In Thomas et al. (2008), the hyperbolic function by which respiration data points are isolated, is referred to as a "hyperbolic deadband". We do not believe that a deadband, which generally describes a range in one dimension, is a logical term when describing a hyperbolic cutoff in 2D.

*Also the acronym ODR for the best fit straight lines, in the caption to Fig 1c, could be explained in section 2.1.*

Thanks for the suggestion, this has been added

- (L.77) Acronym ODR added to explanation.

*p3,5   The definition of an understory ejection as 0.5s with w > 0 and above a regression line, seems a little arbitrary. Were other criteria tried?*

The concept of ejections being somewhat persistent in time made us start with longer 'minimal durations', of tens of seconds. Ultimately, we realised that short intense gusts were eliminated, while they can still be considered ejections, and carried physically relevant information. Our chosen limit of 0.5 s includes such shorter ejections.

*p4      Figure A3 might need more explanation, or at least a forward reference to Figure 3.*

Section 2.2, where the method is described, clarifies (the need for) figure A3 and refers to figure A3. We recognize that inversely, section 2.2 should be referenced in the caption of figure A3. A forward reference to figure 3 (main text) is already present in the caption of figure A3.

Changes:

- (cap Fig. A3) reference to section 2.2 added.

*p7, 9, Fig 3,  Was it a 13 day or a 14 day campaign? Could impact the number of data points averaged in Fig 3. It might also be useful to say how many days had sufficient "frequent understory ejections" in the 30 min time slots. What was the limit for data in Fig 1D - looks like about 6%.*

This is well spotted. The reliable data period used for our analysis runs from August 8th in the morning to August 20th in the evening, so covers 13 full days. Fortunately, our averaging function for deriving composites was not dependent on a manually specified number of days.

As for the ejections, each day contributed at least minor ejections, and we have added this to the manuscript. Larger ejections, of long durations, were more rare.

Fig 1D provides an accumulation of the duration of all these occurrences, in 30 min bins. We did not apply a limit for the data in Fig. 1D.

Changes:

- (L.156, L.209) Change 14 to 13 days.
- (L.143) Specify that all days contributed.

**References**

González-Armas, R., Rikkers, D., Hartogensis, O., Quaresma Dias-Júnior, C., Komiya, S., Pugliese, G., Williams, J., van Asperen, H., a-Guerau de Arellano, J., & de Boer, H. J. (2025). Daytime water and CO2 exchange within and above the Amazon rainforest. *Agricultural and Forest Meteorology*, *Preprint*.

Thomas, C., Martin, J. G., Goeckede, M., Siqueira, M. B., Foken, T., Law, B. E., Loescher, H. W., & Katul, G. (2008). Estimating daytime subcanopy respiration from conditional sampling methods applied to multi-scalar high frequency turbulence time series. *Agricultural and Forest Meteorology*, *148*(8–9), 1210–1229. https://doi.org/10.1016/j.agrformet.2008.03.002

---

## Author Comment (AC2)

Rebuttal of ACP manuscript; **Amazon rainforest ecosystem exchange of $CO_2$ and $H_2O$ through turbulent understory ejections**

We kindly thank the reviewers and the editor for the time they invested in improving the manuscript, and for their appreciative general remarks. We have addressed individual comments in blue, and attached a track changes manuscript to indicate how we incorporated the feedback.

Comments from reviewer 2

*This is an excellent and novel paper focused on using high-frequency turbulence observations to evaluate the dynamic nature of rainforest canopy venting events. It is quite appropriate for the journal and generally of high quality. The datasets are quite valuable and unique and are likely to be used by others. I did not download the data but did verify that the authors have posted the data publicly already.*

*It appears that the authors did not extract xylem water for isotopic analysis (no data are shown). This was an unfortunate oversight for such an isotope-intensive project. For this reason, your claims about knowing the isotopic signature of understory transpiration are rather weak. I highly recommend tuning up the modern Craig-Gordon model (as described by Cernusak et al. 2016) to calculate transpiration signatures based on the data in Fig 3B.*

Xylem water

We agree that optimally, xylem water was sampled in addition to the rootzone, runoff, and leaf water samples. However, xylem water extractions also have their limitations, and are time intensive (de Deurwaerder et al., 2020). Since we study the ecosystem scale, and not individual plants, multiple plants would have needed to be sampled, too. Since resources and time were limited, we took the informed decision to focus on runoff water, and horizontally heterogeneous samples of rootzone water. As far as we are aware, water uptake is (still) understood to be fractionation free, so as long as the measurements are representative, we can approximate the xylem water composition well (Rothfuss & Javaux, 2017).

We are aware that some studies warn that sampling rootzone water to approximate xylem water is challenging (de Deurwaerder et al., 2020; Vega-Grau et al., 2021). Yet, this is true in the context of their studied ecosystems, which are temperate forests with clear wet and dry seasons. Soil evaporation therefore causes strong gradients in rootzone water in these secosystems, from which plants extract a hard to define subset. We however were in a rainforest, with all year water excess. This flushing of the soil with rainwater equilibrates the isotopic gradients.

A second feature of rainforests is that soil evaporation is very small, as little solar energy reaches the surface through the dense canopy (González-Armas et al., 2025).

This prevents strong vertical isotope gradients to emerge. Moreover, horizontal heterogeneity in rootzone water is generally caused by horizontal gradients in soil evaporation, often due to localized shading. No such local shading is present in a rainforest, suggesting that both horizontal and vertical gradients were limited.

We believe the soil sample analysis verifies this rather uniform water isotopic picture. While at other sites vertical gradients of 40 ‰ $\delta D$ are regularly observed (Canet-Martí et al., 2023), we find 4 ‰ $\delta D$ on average. Moreover, the variability between the four sampling points, which were up to 100m apart, was small. The boxplots in figure 3b indicate an Inter Quartile Range of 3 ‰ $\delta D$ for the deep soil water. In comparison, in a temperate environment, de Deurwaerder et al. (2020) find a $\delta D$ variability of 25 ‰ in the xylem water of one tree during one day.

Craig – Gordon model implementation

During the analysis phase of our dataset, a quantitative relationship between the liquid water samples and the vapor source composition was considered but not followed given the scope of this manuscript (understory ejections). In current work for an envisioned follow up manuscript, we analyse the isotopic water and carbon cycles using isotope fluxes for flux partitioning. There, we focus on bulk canopy fluxes only.

Following the comments from reviewer 2, we have now implemented the Craig-Gordon model (Craig & Gordon, 1965) and present the results here. In the line of argumentation of our manuscript, we compare the leaf water $\delta D$ isotopic compositions from the understory and bulk canopy. We combine data from our work with ecophysiology and profile measurements from our colleagues, which were also taken during the CloudRoots Amazon22 campaign (González-Armas et al., 2025). A 14:00 case from a full-campaign data composite was analysed. The key variables derived from either the data composite or the literature are displayed in the table below.

| Variable | Canopy top (bulk, ~25m) | Understory (~5m) |
|---|---|---|
| $\delta D_a$, atmospheric vapor | -65 ‰ | -65 ‰ |
| $\delta D_s$, xylem water approximated from mean of deep soil and runoff water samples | -24 ‰ | -24 ‰ |
| $RH$, relative humidity (in canopy profiles) | 0.58 | 0.65 |
| $T_{leaf}$ (air temp in canopy profiles) | 33 °C | 30 °C |
| $R_s$ (measured stomatal resistance) | 136 s m$^{-1}$ | 512 s m$^{-1}$ |
| $R_b$ (modelled boundary layer resistance, dependent on U, following Bonan (2002)) | 77 s m$^{-1}$ | 245 s m$^{-1}$ |
| $\alpha^+$ (liquid to water fractionation, following Horita and Wesolowski (1994)) | 1.0706 | 1.0735 |
| $\alpha_k$ (kinetic fractionation, following Farquhar et al. (1989)) | 1.0221 | 1.0224 |
| $\delta D_e$, exchange site liquid water (from Eq. A1) | 29.2 ‰ | 27.3 ‰ |
| $\delta D_L$, leaf sample liquid water (avg, see Fig. 3B) | 9.6 ‰ | 7.6 ‰ |

We applied the modern Craig-Gordon model, as is commonly used in plant water isotope studies (Barbour et al., 2017; Cernusak et al., 2016; Farquhar et al., 2007). The model output suggests that the exchange site liquid water was enriched in Deuterium to nearly 30 ‰. This is 20 ‰ higher compared to the composition of the bulk leaf samples. This difference is expected, as the water in the leaf lamina and the veins is a mixture between the source water and exchange site water, regulated by the Peclet effect (Cernusak et al., 2016). Importantly, exchange site liquid water isotopic compositions cannot reliably be converted to a leaf averaged water isotopic composition. Approximations for this link do exist (e.g. eq. 16 of Cernusak et al. (2016)), but these are highly dependent on an uncertain empirical effective path length and assumptions regarding the distribution of water between the veins and the lamina (Farquhar & Lloyd, 1993).

The model output also suggests a 1.9 ‰ stronger enrichment in the leaves from the canopy top, compared to understory leaves. This is qualitatively and quantitatively similar to the observed difference in the leaf samples we took, in which the top canopy leaves were enriched by 2.0 ‰ (note that the enrichment found in the leaf samples is not statistically significant, see comment L.197).

Comparing the output of the Craig – Gordon model to the Miller-Tans derived source isotopic compositions of the bulk canopy (Fig 3 of manuscript) is possible, but not insightful. This is because steady-state models assume the isotopic composition of evaporated water vapor to match the composition of the liquid source water. Formulations for the isotope ratio of evaporated air (RE), like given in Barbour et al. (2017) equation 4, thus literally solve for the source isotopic composition (Rxylem), which we already know from the rootzone and runoff water. The link we find in our study between the bulk (equilibrium) vapor source compositions (-25 ‰) and δ xylem (-24 ‰), is more meaningful, as it is not dependent on steady state assumptions. We already mention this finding in L. 199 - 201, for readers interested in the water cycle.

The Miller-Tans derived understory source composition cannot usefully be compared to the output of the Craig-Gordon model. This is because understory air represents an unknown mix of understory (leaf) transpiration and soil evaporation, while the Craig-Gordon model is only applicable for leaves. We can safely assume that δ evaporation is much more depleted than δ transpiration, as soil evaporation is not an equilibrium process (see Fig. A). Only if the top soil water (-6 ‰) would approach the isotopic composition at the leaf exchange sites (29 ‰) could we expect δ evaporation to match δ transpiration, which is not realistic in most ecosystems, and especially not in rainforests, as described above.

[Figure]

*Figure A: From Dan Yakir's lab website: https://www.weizmann.ac.il/EPS/Yakir/stable-isotopes-lab*

In principle, we could partition the evaporation (E) and transpiration (T) fluxes in the understory to first order. The isotopic compositions of the leaf transpiration and soil evaporation would be used end members, and the understory vapor source composition should then allow us to solve for the relative contributions of E and T. For us, the large uncertainty in the Miller-Tans derived understory vapor source composition is the main reason not to pursue this path. The scale difference between ecosystem scale ejections and local leaves, and need for approximations for e.g. the soil diffusive fractionation, are also important considerations.

We know from our measurements that the understory vapor source is depleted compared to bulk canopy air by about 10 ‰. This depletion is stronger than the measured 2.0 ‰ depletion of liquid water samples from understory leaves compared to top canopy (bulk) leaves. Qualitatively, soil evaporation can be

understood to close this logical gap, as it is known to be much more depleted compared to leaf transpiration.

Conclusion and changes to manuscript

We recognize that the original manuscript addresses the isotopic water cycle too briefly, which leaves specialist readers with questions. We therefore implemented some improvements which should better communicate the coherence of our understanding of the water isotopic state of the plants and the soil.

We appreciate the idea of including a quantitative link between the vapor source compositions, and the leaf and soil samples. Thus, we implemented the Craig – Gorden model to deepen our understanding of the differences between the bulk canopy and the understory. The outcome is in line with our understanding and the results, and we believe it can help the reader understand the water isotopic context. We have therefore added the results to our main text. We find, however, that the CG model does not allow for the isotopic compositions of the liquid water samples and the vapor source compositions derived with the Miller-Tans method to be related quantitatively. As this limits the importance of the CG implementation to the reader, we move it to the appendix.

More elaborate analysis methods include using complex non-steady-state Craig-Gordon models, empirically linking evaporation site enrichments to leaf enrichments, and partitioning the E and T fluxes (in the understory). While we choose not to pursue these now, we hope to apply some of these methods in our upcoming work in which we plan to partition bulk $H_2O$ and $CO_2$ exchange using isotope fluxes.

Changes:

- (L. 197) Added "An ANOVA test indicates that the depletion of understory leaves compared to bulk leaves is not significant for our sample (p=0.34)"
- (L. 198) Added "The Craig - Gordon model was used to determine the isotopic composition of the liquid water at the exchange site ($\delta e$), as a function of the isotopic composition of atmospheric water vapor ($\delta a$), the source water ($\delta s$) of the plant, and environmental variables (appendix A3). In line with our understanding, it suggests that $\delta e$ of leaves from the bulk canopy is 20 ‰ more enriched compared to the leaf water average derived from our sample (Cernusak et al., 2016). In addition, $\delta e$ of understory leaves is depleted by 1.9 ‰ compared to the leaves from the bulk canopy. This supports the idea that understory leaf water is comparatively depleted too."
- (L. 199) replaced "Thus, evaporation of water from understory leaves and the soil explains the lower $\delta D$ source signatures observed for the understory ejections." with "The 10 ‰ Deuterium depletion of understory source air in Fig. 3 cannot qualitatively be explained by a ≈2 ‰ depletion in leaves only.

Liquid water in the soil is about 20 ‰ depleted compared to leaf water (Fig. 3B), and thus evaporated vapor from the soil should have a similar depletion of 20 ‰ compared to transpiration from leaves. We can thus expect that contributions from soil evaporation, which contribute to the understory source air, explain the 10 ‰ depletion we find."

- (L. 200) Added "Plant source water is the average of the water in the root zone, which can be sampled from the plant xylem (de Deurwaerder et al., 2020). We approximate the isotopic composition of the source water by taking the average of the deep soil samples (-20 ‰), and a sample of taken from a stream draining the local plateau (-27.4 ‰ at ≈70 m lower elevation). This approximation is supported by the fact that plant water uptake does not cause isotopic fractionation, and the knowledge that horizontal and vertical water isotopic gradients are limited in a rain forest with excess precipitation (Rothfuss and Javaux, 2017; Vega-Grau et al., 2021)."
- Appendix A3 was added, which details the implementation of the Craig-Gordon model, including the table of variables.

*The introduction is well written and concise, with a clear hypothesis. I have many mostly minor comments that should be addressed before publication. However, this paper definitely should be published! Nice work!*

*It would be great to see a version of Fig 2 for sensible and latent heat fluxes also, and maybe add typical profiles during midday of the various scalars.*

Below we share the new version of Fig 2 and include the $H_2O$ quadrant flux figure.

[Figure]

[Figure]

We find a strong daytime evapotranspiration flux, contrasted by relatively small nighttime fluxes. In the early morning, some negative moisture fluxes – potentially indicating dewfall - are present, but of a small order of magnitude (-0.04 mmol m$^{-2}$ s$^{-1}$). As the quadrants are defined in $H_2O$-$CO_2$ space, the partitioning per quadrant is comparable to the $CO_2$ quadrant flux plot. Q2 and Q4 represent the dominant exchange mode during daytime, while Q1 and Q3 relate to nighttime and respiration rich episodes.

During our analysis phase, we already spent time investigating the $H_2O$ quadrant fluxes. We find that the (effectively) unidirectional fluxes in all quadrants limit the additional value of partitioning the fluxes in quadrants. Whilst we find the idea of a quadrant investigation into the temperature fluctuations interesting, we consider it to be outside of the scope of this study.

The main comment of reviewer 1 was about profiles and potentially linking them to observed ejections. As the response covers two pages, we (only this time) refer you to that rebuttal to see how we addressed in-canopy profiles. In short, we share profile data measured at the site during our campaign. While a stable layer with a buildup with $CO_2$ is clearly observed, we do not see viable ways of linking both directly, due to the scale differences.

Changes:

- We include the $H_2O$ quadrant flux figure in the appendix
- We add references to the appendix in Fig. 2, L.72, and L.177
- (L.45)
  - We refer to the work of (González-Armas et al., 2025)

- o We specify the ecosystem scale of our measurements and indicate that direct links to profiles were not considered insightful.
  - (L.240 (discussion))
    - o We mention the temporally averaged local profiles taken during CloudRoots Amazon22 as presented in Gonzales-Armas et al (2025), which confirm a stable layer in the canopy with a visible respiratory signature.

*Abstract: this statement is incorrect "We show that this matches the depleted water vapor isotopic compositions found in understory leaf and soil samples".  Understory leaf water was about +8 ‰ $\delta D$, and soil water was -15 to -20 ‰ (Fig 3B).  All of these values are more enriched than the background atmosphere and the estimated values of source composition (Fig 3A).  Where is the match?*

This point was also raised by referee 1, and we acknowledge that this statement is misleading. Specifically, the word "vapor", suggests that understory leaf and soil water vapor was measured, while only the liquid water was measured. The match we suggest is a qualitative one. Namely, that understory leaf and soil liquid water are depleted compared to bulk leaf liquid water. A logical consequence is that the vapor from these depleted liquid sources is also more depleted than the vapor transpired from enriched leaves.

Changes:

- (L. 8) Remove the word 'vapor' and specify that liquid understory water was comparatively depleted compared to liquid 'bulk' water.
- (L. 199) replaced "Thus, evaporation of water from understory leaves and the soil explains the lower $\delta D$ source signatures observed for the understory ejections." with "The 10 ‰ Deuterium depletion of understory source air in Fig. 3 cannot qualitatively be explained by a ≈2 ‰ depletion in leaves only. Liquid water in the soil is about 20 ‰ depleted compared to leaf water (Fig. 3B), and thus evaporated vapor from the soil should have a similar depletion of 20 ‰ compared to transpiration from leaves. We can thus expect that contributions from soil evaporation, which are also part of the understory source air, explain the 10 ‰ depletion we find."

*Minor comments:*

*3: forest atmosphere interface (not interphase)*

Adjusted accordingly

*12: weak but coherent (r=0.027) - unclear meaning upon reading abstract*

We agree that this is unclear for a new reader. We struggled with this formulation and ended up with a too nuanced statement for an abstract. The key takeaway is that a weak relationship was found.

Adjusted accordingly

*14-15: how can a deeper understanding of gas exchange prevent the transition from C sink to source?*

This is a fair point. The logical pathway – as is true for most climate research – would be general climate change mitigation policy. Reliable biosphere feedback predictions do clarify the non-linear consequences of future emissions. That knowledge does allow policy makers to take into account both the effect on the rainforest, and of the extra emissions from the rainforest. Compared to not knowing, knowing thus gives the rainforest a larger chance of remaining a sink.

Changes:

- (L.14) adjusted "is urgent for predicting and possibly preventing" to "which may ultimately allow decision makers to incorporate policies which can prevent"

*48: 20 Hz sampling rate here, but Fig 1 caption says 10 Hz*

Sonic and OPGA data were taken at 20 Hz, while Fig 1 indeed displays 10Hz data. This subsampling was needed to link the EC measurements to the isotope analysers ($H_2O$-iso at 4Hz, $CO_2$-iso at 10Hz). We performed our analysis on both these 10Hz and 4Hz 'aligned' datasets. In this work we do not use the $CO_2$ isotope data. Still, we used that data for Fig 1, as it has a higher frequency. Making Fig 1 with 20Hz data instead would require redoing our analysis for that data set, and is not expected to add value.

Changes:

- (L.54) The subsampling of EC data dependent on the isotope analyser measurement frequency is now specified.

*60: improved and improvement redundant*

Adjusted accordingly: removed improved

*74: other usage of the hyperbolic isolation function for isotopic exchange can be found here: Bowling et al (1999) JGR 104:9121, and (2003) AFM 116:159*

Adjusted accordingly: Reference to Bowling et al (1999) added.

*76: for isolation OF wind sweeps and ejections*

Adjusted accordingly

*79: 2 sigma represents standard deviation of the residuals from the ODR fit? - more mathematical detail needed here so others can reproduce this*

Correct, the standard deviation of the residual (y) is meant. Note that the residuals in x, or a std defined perpendicular to the fitted line, would all result in the same effective threshold due to symmetry.

Changes:

- Specify that the std of the residuals was used
- (L.80) Clarify 2.5 sigma boundary.

*80: ejections are Q1 not Q2 according to your notation in Fig 1 (your notation is consistent with Thomas et al. (2008) and those events were identified by them as subcanopy-venting updrafts - please be clear that the labeling of quadrants is dependent on the variables plotted - the Shaw ejections in Q2 (your Fig A5) are not the same meaning as Q2 in your Fig 1*

This is well spotted and correct. Indeed the ejections in Fig 1 are in a different quadrant compared to the wind ejections in Fig A5 (after Shaw et al. (1983)). For that reason, we had used an inverted co2 (x) axis in an earlier version of this manuscript. The quandrant labeling was a leftover from that version and is now corrected.

Changes:

- We have substituted Q2 with Q1
- (L.67) Note the dissimilarity between quadrants in $CO_2$-$H_2O$ and u–w space.

*69-72: more detail would be helpful here - it seems you are splitting the total turbulent flux based on the quadrants, and for each quadrant taking a Reynolds average of the instantaneous product of w and scalars (or w and u)? so total turbulent flux = sum of turbulent fluxes in each quadrant?*

The total flux was not split, but the fluxes were calculated seperately for each quadrant. The sum then matches the total flux addording to:

$$F_{tot} = \sum_{i=1}^{4} F_{Qi} = \rho_m \sum_{i=1}^{4} \overline{w'_{Qi} \chi'_{Qi}} \frac{n_{Qi}}{n_{tot}}$$ Eq. 1

This equation has been added to the revised manuscript

Changes:

- (L.68) Elaborate on approach for calculating the quadrant fluxes
- (L.72) Add eq. 1., including a description of the variables.

*80-83: more detail needed here - show us the equations that define how you calculate the bulk and ejection fluxes*

Adjusted accordingly: Clarify formulation and link directly to eq. 1. The fluxes calculated with the subset 'bulk', and the subset 'ejection', now sum to the total flux.

*87: are those 36 consecutive points or 36 from anywhere within the 7200?*

We mean 36 points from anywhere. We believe that the formulation does not suggest consecutive points. Moreover, adding a specification of 'anywhere' or 'random' is believed to decrease the clarity of the formulation.

*88: one-sided uncertainties is vague, and more detail would be helpful on why "unrealistic source compositions" occurred*

In Miller-Tans equations, the value of the slope represents the source isotopic composition. The standard error of the slope of the fit was derived following York (1968). Here, the uncertainty in the input variables is propagated to the uncertainty in the slope. The precision of the 4Hz $H_2O$ molefraction data was ~1ppm, and the precision of the 4Hz dD was ~0.5 ‰. Ultimately, we used a limit of 6 ‰ to exclude extreme outliers (and not 9 ‰, which was stated in the original manuscript).

The outliers in the source isotopic composition which were deemed unrealistic are difficult to understand. Most unrealistic values occurred during nighttime. After looking into some examples, we suspect that changes in air masses, possibly related to nighttime atmospheric hetrogeneity, play a role. Here, the one-source mixing assumptions underlying the Miller Tans (and Keeling) methods are likely violated.

Changes:

- We clarify the error propagation through Miller Tans slope fit.
- We explain that standard errors in the slope of more then 6 ‰ were discarded.
- (L.89) we clarify that the source mixing assumptions underlying the Miller-Tans approach were likely violated at times.

*95: incomplete sentence*

Adjusted accordingly: finished logically

*108-112: this is too vague to understand - this looks like a time-lagged cross correlation between "understory ejection intensity" and "cloud intensity"??? what are "two time series" and why did you not look for the time of maximum cross-correlation? - it seems that Fig 4 D should be labeled to be consistent with these variables (is "ejection occurrence" the same as "understory ejection intensity"?*

We have improved the description, implementing the suggested changes, also to the labels in figure 4 (A!). The time of maximum cross-correlation is indeed relevant, and specified in section 3.4 and Fig 4B.

Changes:

- We use the term time-lagged cross correlation (in the methods and in the resutls) instead of describing the method.
- We immediately specify the "cloud intensity" and "understory ejection intensity", instead of referring to these as 'the time series defined in the paragraph before'.
- We change the labels of Fig 4A to "cloud intensity" and "understory ejection intensity".

*114: closest cloud onset in time?*

Adjusted accordingly: Note that in our analysis we only have a time dimension.

*Fig 2: please use molar units for the fluxes as in Fig 1*

Adjusted accordingly

*Fig 2 caption: last sentence is vague please provide detail*

Adjusted accordingly: we refer to the newly added Eq. 1, which specifies the method.

*122: associated instantaneous vertical wind speed*

Adjusted accordingly

*125: "anti-correlation between photosynthesis and transpiration" - the wording could be improved, these processes happen together through the stomata - you mean the fluxes go in different directions vertically (and thus their mole fractions in the atmosphere in the presence of these fluxes are anti-correlated)*

Adjusted accordingly: anti-correlated is now related to the mole fractions. Assimilation and transpiration fluxes are instead described as opposed.

*129: I don't understand why you refer to Q3 as "stable background" when the entire example is stable stratification - why only Q3 and not the others?*

We intended to communicate that respiration and evapotransporation are mixed into the atmospheric reservoir, which is more stable in composition, and shown in Q3 (negative w'). We agree that stable in this context has a double meaning, and can thus better not be used.

Change:

- Replaced "stable background" with "atmospheric background"

*Fig A1 in appendix: conflicting terminology here, "Q1 Respiration" (left column), but also right column says "understory ejection", but the caption says ejections are in Q1*

We have checked the terminology again and think that it is not conflicting. Daytime understory ejections indeed have a respiration signature. Therefore there is overlap between the fingerprints of Q1 and the understory ejections. Indeed, as shown in Fig. 1C, the ejections predominantly take place in Q1.

*141: seems like more than 2 events in Fig A1 rightmost column - at least 3 smaller ones there too*

Adjusted accordingly: we clarify that the ejection in Fig 1C "mostly" represent the two updrafts.

*148: "The bulk of the flushing takes place from 7:00 to 9:00 LT, but the positive relationship between the H2O and CO2 anomalies prevents ejection events from being isolated then" - this is not what I see in that figure - there are ejection events identified prior to 09:00, and the "two phases" of ejections mentioned are not very distinct in the histogram (Fig 1D). Ejections seem to occur throughout the day and are not broken into two clear groups*

We agree that Fig 1D does not allow for the two ejection phases to be clearly seperated. Still, there is a notable ejection-lean period at 10:30. In the text, we therefore refer to Fig. 2. There, the initial period of ejections is clearly related to the early morning flushing (Q1,Q3), while the period from 11:00 is not. Importantly, the flushing after 9:00 is more likely to be observed as an ejection, as the bulk flux has turned to photosynthesis-dominant by then. According to both Fig. 2b and Fig. 1D, hardly any ejections (and flux contributions) occur before 9:00.

*151: are there observations of CBL height to compare to strengthen this argument?*

Due to the canopy at ATTO, there are no profiling instruments located there. At the nearby Campina site, some profiling instruments are available. However, analysing

the boundary layer development is not in the scope of this study. Linking our findings to general boundary layer behaviour thus seems sufficient to us.

Change:

- Added a reference to the chapter Atmospheric Boundary Layers of the book Fundamentals of Meteorology from Spiridonov & Ćurić (2021).

*Section 3.2 title: "CO2 flux partitioning" means splitting NEE into GPP and Reco - consider rewording to something like "quadrant analysis of CO2 fluxes"*

Adjusted accordingly

*156: Fig 2 has only 1 panel so no "A" needed*

Adjusted accordingly

*158: "sporadically during noon" - this is not my interpretation of Fig 2, I see sporadic events from 0900 to 1800*

We agree. Our intention was to refer to the daytime period around noon by using "noon", which we realise refers in English language only to 12:00 sharp.

Change:

- We specify the time window from 09:00 to 17:00.

*159: I don't understand the 3.8 to 20% numbers.  The largest ejection flux in the figure has a magnitude of 0.01 mg m-2 s-1, but the net uptake of that time was ~-0.38, so the ejection flux contributes 100x 0.01/0.38 = 2.6%.  Please explain your calculations in detail*

*240: more detail needed on the 1.4% - how did you calculate this?*

We derived these three numbers from the entire 13 day dataset underlying Figure 2. In that dataset, when ejections occur in a 30 minute interval, they average 3.8 % of the total $CO_2$ flux. Contributions up to 20% are observed also, but that is the high end of the distribution. When concidering all 30 min intervals, including those without ejections, ejections contribute 1.4% of the NEE flux.

Change:

- We restructured the paragraph to more clearly indicate the difference between the 3 numbers.
- We clarify that these statistics are taken from the full dataset of 13 days, and not from Fig 2.

*183: Fig A3 does not show either Keeling or Miller-Tans plots - why even mention Keeling plots if you didn't use them?*

We intended to reference Fig A4 and not A3. This has now been solved. In Fig A4, both Keeling and Miller-Tans plots are shown. We believe it is good to show Keeling plots as they are still much more commonly used, and more intuitive to understand.

Changed:

- Fig. A3 → Fig. A4

*197: "shaded understory leaves and soil evaporation are isotopically depleted compared to bulk sunlit leaves" I see 2 problems here. Most serious is that you have not measured the isotopic composition of soil \*evaporation\*, but bulk soil water. Second, ANOVA is needed to make the claim that understory leaves are more depleted than bulk sunlit leaves. Using the Craig-Gordon model would strengthen these claims*

As presented in the rebute of the main comment of reviewer 2, we have now in more detail adressed the water isotopic cycle. Here, we used the Craig - Gorden model, and clarify that while we have not measured the isotopic signature of soil evaporation, we can approximate it from the top soil water, assuming equilibrium fractionation.

The results of an ANOVA between the leaf bulk leaf samples and those taken from the understory results in a P-value of 0.34 (and an F-value of 0.92). We can thus not say that the understory leaves are significantly more depleted then bulk leaves are.

Change:

- (L. 197) Changed "soil evaporation", to "top soil water from which evaporation takes place".
- (L. 198) Added "An ANOVA test indicates that the depletion of understory leaves compared to bulk leaves is not significant for our sample (p=0.34)"

*204: "wind ejections" is used here to refer to understory ejections, but earlier when discussing Shaw's paper you make a distinction between these - better to leave out "wind" here*

The "wind ejections" mentioned in line 204 are intended to refer to the u'w' ejections from Shaw et al. (1983). So, while reviewer 2 thought we meant to refer to understory ejections, we specifically wanted to indicate u'w' based ejections with the word 'wind' ejections. No changes were made based on this comment.

*210-213: this is too vague to understand*

In the discussion (247-251), we explain our investigation into wind – ejections relations ships more clearly. There, we focussed on the high frequency wind – ejection link. Now we also mention the results of Fig. A6, where we test if periods with ejections are statistically more gusty.

Changes:

- We remove the unclear section wind – ejection relationships from the results (L.210
- (L.251) We expand on our understanding of the wind – ejection relationship in the discussion, including the lower frequency wind gust analysis.

*214: I would refer to this as time-lagged cross correlation (https://en.wikipedia.org/wiki/Cross-correlation)*

Adjusted accordingly: Terms changed in both the methods and results

*215: I am confused about the distinction between cloud onset (start of shading) and your use of "center of clouds" here. Fig 4C makes me think that ejections happen before the cloud arrives, but you are making the opposite point I think (clouds lead to ejections). More clarity needed here*

We mention both 'cloud onsets' and 'cloud centers' explicitly, as we perform an anlysis with both of these variables. The time-lagged cross correlation (Fig. 4B) naturally indicates the averaged distance of ejections to the the center of clouds, while the ejection distribution (Fig. 4C) was plotted around the cloud onset. In L. 216 we mention that the cloud center was 'selected' when discussing the time-lagged cross correlation, while it is rather a concequence of the method. We suspect that caused some unclarity for reviewer 2. Moreover, we find that in line 231, we accidentally refer to the cloud center, while we intended to say cloud onset.

Indeed we suggest that clouds might be both a trigger for ejections (although the correlation is low), and that ejections preceed the arrival of the cloud center. We speculate that either the initial shading of cloud onsets, or the wind dynamics around (and 'ahead of') the cloud are ejection triggers. We more clearly indicate this by adding a sentence to the description of Fig. 4C, which shows the preferential presence of ejections around cloud onsets.

Changes:

- (L.216) We clarify that time-lagged cross correlation naturally takes the cloud center as the anchor point.
- (L.231) We correct a mistake in using 'cloud center' instead of 'cloud onset'
- (L.235) We indicate that Fig. 4C supports the idea that ejections are related to cloud onsets, rather than cloud centers.

*216: radiation (not radiate)*

Adjusted accordingly

*Fig 4C ejection is spelled wrong in the legend*

Adjusted accordingly

*221: the very low correlation coefficients worry me, and really weaken your argument that these are correlated - can you do more simulations to see if these values of r can be achieved randomly?  maybe randomize both time series independently many times then repeat the test and compare the results*

We agree that the correlation coefficients are low, and lower than what we expected based on our hypothesis, as indicated in the discussion and conclusion. As we do not state that clouds and ejections are clearly correlated, we are comfortable in sharing this weak correlation. We would be more hesitant if the pattern in the time-lagged cross correlation was erratic.

We found it more natural to apply simulations as suggested by reviewer 2 in analysis 2 (Fig. 4C). There we randomly vary the cloud fields to prove the statistical robustness of our findings. The result indicates that the likelyhood of ejections occuring near a cloud onsets exceeds what would randomly be expected. That confirms that there is likely some link between clouds and ejections.

*232: the synthetic simulations need more detail, I don't understand*

To find out how our result compares to cases where clouds and ejections certainly are not linked, we created 379 'synthetic' cumulative histograms. We did this by applying a time shift to the cloud field of at least 1 hour compared to the real measurement, and differing by 15 minutes between synthetic samples. Ultimately, each ejection gets colocated with 379 unrelated cloud fields. The advantage over designing cloud or ejection fields manually or randomly is that here, the temporal distribution between clouds remains realistic. Only data from the 09:00 – 16:30 period, during wich ejections are present, were used. The synthetic data in the figure is the average histograms over these 379 runs.

Changes:

- (L.232) we add details on how the synthetic distribution was made.
- (L.233) we better describe how the observations deviate from the synthetic case.

*241: moist air yes, but you do not show evidence for saturation of water vapor in air*

Adjusted accordingly: saturated replaced by moist

*242: "depleted" is a relative term.  Understory ejections seem to have δD values of -30 to -40 ‰, which is \*enriched\* relative to the atmospheric background (Fig 3A).  The understory transpiration flux is likely close to deep soil water (-20 ‰) which is also very enriched relative to background.  The Craig-Gordon model would help here.*

We have now described the isotopic story in more detail, and included the Craig-Gordon model. We have specified in L.242 that the depletion is with respect to the vapor from the bulk canopy transpiration.

Changes:

- Specify depletion with respect to bulk canopy transpiration.

*252-261: this is a wandering paragraph and rather speculative*

We decided to keep the paragraph in as we believe the conciderations we share put our analysis in context, and can help future experimental design.  For clarity sake, we leave out the suggested translation from 1D temporal space to 1D physical space. In addition, we more clearly separate the effect of wind speed (which does not matter for our annalysis) and wind shear (which does introduce a bias).

Changes:

- Remove L. 260
- (L. 258) clarify that strong wind shear might cause uncertainties, not high wind speeds.

**References**

Barbour, M. M., Farquhar, G. D., & Buckley, T. N. (2017). Leaf water stable isotopes and water transport outside the xylem. In *Plant Cell and Environment* (Vol. 40, Issue 6, pp. 914–920). Blackwell Publishing Ltd. https://doi.org/10.1111/pce.12845

Canet-Martí, A., Morales-Santos, A., Nolz, R., Langergraber, G., & Stumpp, C. (2023). Quantification of water fluxes and soil water balance in agricultural fields under different tillage and irrigation systems using water stable isotopes. *Soil and Tillage Research*, *231*. https://doi.org/10.1016/j.still.2023.105732

Cernusak, L. A., Barbour, M. M., Arndt, S. K., Cheesman, A. W., English, N. B., Feild, T. S., Helliker, B. R., Holloway-Phillips, M. M., Holtum, J. A. M., Kahmen, A., Mcinerney, F. A., Munksgaard, N. C., Simonin, K. A., Song, X., Stuart-Williams, H., West, J. B., & Farquhar, G. D. (2016). Stable isotopes in leaf water of terrestrial plants. *Plant Cell and Environment*, *39*(5), 1087–1102. https://doi.org/10.1111/pce.12703

Craig, H., & Gordon, L. I. (1965). *Stable Isotopes in Oceanographic Studies*.

de Deurwaerder, H. P. T., Visser, M. D., Detto, M., Boeckx, P., Meunier, F., Kuehnhammer, K., Magh, R. K., Marshall, J. D., Wang, L., Zhao, L., & Verbeeck, H. (2020). Causes and consequences of pronounced variation in the isotope composition of plant xylem water. *Biogeosciences*, *17*(19), 4853–4870. https://doi.org/10.5194/bg-17-4853-2020

Farquhar, G. D., Cernusak, L. A., & Barnes, B. (2007). Heavy water fractionation during transpiration. In *Plant Physiology* (Vol. 143, Issue 1, pp. 11–18). American Society of Plant Biologists. https://doi.org/10.1104/pp.106.093278

Farquhar, G. D., & Lloyd, J. (1993). Carbon and Oxygen Isotope Effects in the Exchange of Carbon Dioxide between Terrestrial Plants and the Atmosphere. In *Stable Isotopes and Plant Carbon-water Relations* (pp. 47–70). Academic Press. https://doi.org/10.1016/b978-0-08-091801-3.50011-8

González-Armas, R., Rikkers, D., Hartogensis, O., Quaresma Dias-Júnior, C., Komiya, S., Pugliese, G., Williams, J., van Asperen, H., a-Guerau de Arellano, J., & de Boer, H. J. (2025). Daytime water and CO2 exchange within and above the Amazon rainforest. *Agricultural and Forest Meteorology*, *Preprint*.

Rothfuss, Y., & Javaux, M. (2017). Reviews and syntheses: Isotopic approaches to quantify root water uptake: A review and comparison of methods. *Biogeosciences*, *14*(8), 2199–2224. https://doi.org/10.5194/bg-14-2199-2017

Shaw, R. H., Tavangar, J., & Ward, D. P. (1983). Structure of the Renolds Stress in a Canopy Layer. *Journal of Climate and Applied Meteorology*, *22*, 1922–1931.

Spiridonov, V., & Ćurić, M. (2021). Atmospheric Boundary Layer (ABL). In *Fundamentals of Meteorology* (pp. 219–228). Springer International Publishing. https://doi.org/10.1007/978-3-030-52655-9_14

Vega-Grau, A. M., McDonnell, J., Schmidt, S., Annandale, M., & Herbohn, J. (2021). Isotopic fractionation from deep roots to tall shoots: A forensic analysis of xylem water isotope composition in mature tropical savanna trees. *Science of the Total Environment*, *795*. https://doi.org/10.1016/j.scitotenv.2021.148675

York, D. (1968). Least squares fitting of a straight line with correlated errors. *Earth and Planetary Science Letters*, *5*(C), 320–324. https://doi.org/10.1016/S0012-821X(68)80059-7